# Diverse MR1-restricted T cells in mice and humans

Hui-Fern Koay [1,2,6], Nicholas A. Gherardin[1,2,6], Calvin Xu[1], Rebecca Seneviratna[1,2], Zhe Zhao[1], Zhenjun Chen [1], David P. Fairlie [3,4], James McCluskey [1], Daniel G. Pellicci[1,2,5], Adam P. Uldrich[1,2] & Dale I. Godfrey [1,2]

Mucosal-associated invariant T (MAIT) cells express an invariant TRAV1/TRAJ33 TCR-α chain and are restricted to the MHC-I-like molecule, MR1. Whether MAIT cell development depends on this invariant TCR-α chain is unclear. Here we generate *Traj33*-deficient mice and show that they are highly depleted of MAIT cells; however, a residual population remains and can respond to exogenous antigen in vitro or pulmonary *Legionella* challenge in vivo. These residual cells include some that express *Trav1*+ TCRs with conservative *Traj*-gene substitutions, and others that express *Trav1*- TCRs with a broad range of *Traj* genes. We further report that human TRAV1-2- MR1-restricted T cells contain both MAIT-like and non-MAIT-like cells, as judged by their TCR repertoire, antigen reactivity and phenotypic features. These include a MAIT-like population that expresses a public, canonical TRAV36+ TRBV28+ TCR. Our findings highlight the TCR diversity and the resulting potential impact on antigen recognition by MR1-restricted T cells.

[1] Department of Microbiology & Immunology, Peter Doherty Institute for Infection and Immunity, University of Melbourne, Melbourne, VIC 3000, Australia. [2] Australian Research Council Centre of Excellence in Advanced Molecular Imaging, University of Melbourne, Parkville, VIC 3010, Australia. [3] Division of Chemistry & Structural Biology, Institute for Molecular Bioscience, The University of Queensland, Brisbane, QLD 4072, Australia. [4] Australian Research Council Centre of Excellence in Advanced Molecular Imaging, The University of Queensland, Brisbane, QLD 4072, Australia. [5] Murdoch Children's Research Institute, Parkville, VIC 3052, Australia. [6] These authors contributed equally: Hui-Fern Koay, Nicholas A. Gherardin. Correspondence and requests for materials should be addressed to D.I.G. (email: godfrey@unimelb.edu.au)

Mucosal-associated invariant T (MAIT) cells are unconventional T cells with innate-like antimicrobial activity[1,2]. MAIT cells are highly abundant in humans, representing approximately 3–5% of human blood T cells[3], and even higher frequency in other tissues, such as liver where they are up to 40% of T cells[4,5]. MAIT cells have been implicated in immunity to a range of bacterial and viral infections, cancers, inflammatory and autoimmune diseases (reviewed in ref. [6]) although their mechanisms of action and antigenic targets in non-microbial diseases are not well understood.

MAIT cells are typically defined by their expression of an invariant T cell receptor (TCR)-α chain[7]. In humans, this consists of TRAV1-2 joined to TRAJ33[8,9], TRAJ12 or TRAJ20[10,11] with little to no $n$ nucleotide additions at the TCR-α complementary determining region 3 (CDR3α) junction[9]. This pairs with a TCR-β repertoire highly biased toward TRBV6 family members and TRBV20-1[9,10]. This unique TCR has been highly conserved throughout mammalian evolution, suggesting an important and non-redundant physiological role for MAIT cells[9]. Indeed, MAIT cells in mice express an orthologous TCR-α chain consisting of TRAV1 and TRAJ33, which typically pairs with TRBV13+ and TRBV19+ TCR-β chains[9]. In contrast to humans, however, MAIT cells are rarer in mice where they typically form <1% of all T cells, although in some tissues, such as lung, lamina propria and lymph node, they can constitute up to 5% of T cells[12]. Nonetheless, upon antigenic stimulation in vitro[12] or in vivo[2,13], MAIT cells can undergo marked expansion to represent up to ≥50% of T cells. Thus microbial exposure may be an important factor in dictating mature MAIT cell frequencies.

The highly conserved MAIT TCR restricts MAIT cells to the recognition of the major histocompatibility class (MHC) class I-related protein MR1[14]. Unlike classical MHC I molecules whose shallow antigen (Ag)-binding cleft is apt to bind short peptide Ags for surface presentation to conventional CD8+ T cells, the Ag-binding cleft of MR1 includes a small Ag-binding pocket (the A′ pocket) lined with aromatic amino acid side chains, imbuing an ability to capture and present small metabolite compounds for surveillance by the MAIT TCR[15,16]. Like the MAIT TCR, MR1 is highly evolutionarily conserved with approximately 90% sequence homology between the MR1 α1 and α2 domains of humans and mice[17], further suggesting an important physiological role for the MAIT TCR–MR1 axis.

Several MR1-bound Ags have been described[18], including a range of microbial-derived vitamin B2 (riboflavin) derivatives that are antigenic for MAIT cells, such as the ribityl-lumazines 7-hydoxy-6-methyl-8-D-ribityllumazine (RL-6-Me-7-OH) and 6,7-dimethyl-8-D-ribityllumazine (RL-6,7-diMe),[15] as well as the highly potent pyrimidine Ags such as 5-OP-RU[16]. More recently, acetylated RL-6-Me-7-OH, the photolumazines 6-(2-carboxyethyl)-7-hydroxy-8-ribityllumazine (photolumazine I; PLI), 6-(1H-indol-3-yl)-7-hydroxy-8-ribityllumazine (photolumazine III; PLIII), the riboflavin analogue 7,8-didemethyl-8-hydroxy-5-deazariboflavin (FO) and riboflavin itself have been described as MR1-binding ligands[19], although riboflavin and FO were inhibitors rather than activators of MAIT cells. The activating Ags are detected by the conserved MAIT TCR with pattern-recognition-like conformity, where the CDR1α, CDR2α and CDR2β loops straddle the α2 and α1 helices of MR1, respectively, positioning the germline-encoded CDR3α at the apex of the A′ pocket, ready for recognition of the ribityl tail, that is common to the riboflavin-derivative Ags. This key interaction is mediated by a conserved TRAJ33/12/20-encoded tyrosine at position 95 (Tyr95α) and mutation of this residue abrogates reactivity[20–22].

MR1 can also capture vitamin B9 (folate)-derivative, pterin-based molecules including 6-formyl pterin (6-FP)[15] and its synthetic analogue Acetyl (Ac)-6-FP[21]. When bound to MR1,

these ligands are buried deep within the A′ pocket[15,21] and are generally not recognised by the MAIT TCR[20,21]. More recently, a study used in silico docking, in vitro cellular assays and X-ray crystallography to identify a broad range of chemically diverse drugs and drug-like metabolites that can also bind MR1[23]. This included aspirin analogues 3- and 5-formylsalicylic acids, a methotrexate derivative 2,4-diamino-6-formylpteridine (2,4-DA-6-FP) and the anti-inflammatory drug diclofenac[23]. Accordingly, the Ag-binding cleft of MR1 exhibits sufficient plasticity to capture and present a diverse range of small molecules. Despite their ability to bind MR1, most non-ribityl compounds discovered to date do not activate MAIT cells at a population level. Nonetheless, discrete subsets of MAIT cells—as determined by sequence variation at the hypervariable CDR3β loop that sits adjacent to the CDR3α loop at the opening of the A′ pocket—have been shown to recognise some of these Ags, including 6-FP, Ac-6-FP[21,24] and diclofenac[23]. Thus CDR3β hypervariability provides a mechanism for discrete subsets of MAIT cells to discriminate between different Ags.

Beyond MAIT cells, recent evidence suggests the existence of atypical populations of MR1-restricted αβ T cells with diverse TCRs and Ag specificities[7]. Co-staining of human peripheral blood mononuclear cells (PBMCs) with MR1-Ag tetramers and antibodies against the TRAV1-2 segment of the MAIT TCR revealed a diverse population of CD8+ T cells that exhibited subpopulations of MR1-5-OP-RU-reactive, MR1-Ac-6-FP-reactive and MR1-autoreactive cells[24]. Likewise, in vitro MR1-restricted antimicrobial activity was used to isolate TRAV1-2− T cells with diverse TCRs[25]. Interestingly, one of these clones appeared to react to Streptococcus pyogenes, a bacterial species not known to encode the riboflavin synthesis pathway, thereby suggesting recognition of a non-ribityl Ag. Recently, TRAV1-2− T cells that reacted against MR1-overexpressing tumour cell lines in the absence of a foreign Ag have been described[26]. These cells had diverse phenotypic features, distinct from that of MAIT cells. Collectively, these studies suggest the existence of diverse MR1-restricted T cells with broader Ag specificity and unique roles for these cells in both microbial and non-microbial immunity. However, key questions about this axis remain. What is the extent of TCR and Ag diversity in the broader MR1-restricted T cell repertoire? How are atypical TRAV1-2− MR1-restricted T cells developmentally, phenotypically and functionally related to classical TRAV1-2+ MAIT cells? In mice, even less is known about the diversity of MR1-restricted T cells. Indeed, TCR sequencing studies have suggested that mouse MAIT cells all express the invariant TRAV1/TRAJ33 TCR-α chain[12]. Given the high conservation of the MAIT TCR–MR1 axis between humans and mice, it is important to understand whether a similar, broad family of MR1-restricted T cells has been conserved across species.

In this study, we use MR1-Ag tetramers to investigate the TRAV1/TRAJ33− MR1-restricted αβ T cell compartment in mice and humans. We develop Traj33 gene-deleted mice and show that these mice retain a residual population of MR1-restricted T cells expressing a range of TCR-α chain genes including TRAV1+ cells with conservative TRAJ substitutions and TRAV1− cells with diverse TRAJ usage. Furthermore, we identify three distinct populations of TRAV1-2− MR1-restricted T cells in humans that differ in TCR repertoire and phenotypic features. Taken together, this study highlights that MR1-restricted T cells extend beyond classical TRAV1/TRAJ33+ MAIT cells in both mice and humans. This has important implications for our understanding of the scope of Ags that may be recognised by these cells and the function of the broader family of MR1-restricted T cells in the immune system.

## Results

**Generation of *Traj33*-deficient mice.** In order to investigate MAIT cell dependence on TRAJ33 and to generate a mouse line that lacked MAIT cells but retained MR1, we generated *Traj33* knockout (KO) mice by CRISPR-Cas9 mediated gene deletion[27] (Supplementary Fig. 1A). *Traj33+/−* mice were backcrossed to in-house wild-type (WT) C57BL/6 mice for 2 generations and then intercrossed to generate *Traj33+/+* WT, *Traj33+/−* heterozygous (het) and *Traj33−/−* homozygous KO littermates. These were tested for heterozygosity or homozygosity using PCR (Supplementary Fig. 1B) and showed the expected Mendelian inheritance ratio. As it was previously reported[28] that genetic deletion of *Traj18* via PGK-neo^r cassette insertion to generate *Traj18−/−* mice[29] inadvertently disrupted TCR rearrangements using genes encoding Jα regions upstream of *Traj18*[28], we examined the *Traj* usage in CD4+CD8+ double positive (DP) thymocytes from *Traj33−/−* mice to determine whether this deletion impacted on other *Traj* gene usage. We used single-cell sequencing for TCRα usage on DP thymocyte clones from WT and *Traj33−/−* mice. Of the 92 sequences from WT DP thymocytes, we detected 63 cells rearranging *Traj* segments downstream of *Traj33* (*Traj2-32*) and 33 upstream of *Traj33* (*Traj34-58*). Analysis of 139 sequences from *Traj33−/−* DP thymocytes revealed 79 cells incorporating *Traj* segments downstream of *Traj33* and 60 upstream of *Traj33*, indicating that there are no defects in the rearrangement of *Traj* genes upstream of *Traj33* in *Traj33−/−* thymocytes (Supplementary Fig. 2). Furthermore, we also examined NKT cells and γδT cells in *Traj33* het and KO mouse organs and determined that their frequencies and absolute numbers were similar to *Traj33* WT organs (Supplementary Fig. 3A, B).

**Identification of mouse atypical MR1-reactive T cells.** We then compared the presence of MR1-5-OP-RU tetramer+ αβ TCR+ cells between *Traj33* WT, het and KO mouse organs, including thymus, spleen, inguinal lymph nodes, lung and liver (Fig. 1a, b). Specificity of staining was determined using MR1-Ac-6-FP tetramer (Fig. 1a). After pre-exclusion of B cells via electronic gating, these data showed, as expected, that MR1-5-OP-RU tetramer+ TCRβ+ cells were markedly reduced in all *Traj33* KO tissues. There was also a trend towards lower MR1 tetramer+ cells in *Traj33* het mice compared to WT mice, which was statistically significant in the spleen and lung (Fig. 1a, b).

Interestingly, in *Traj33* KO mice, the population of MR1-5-OP-RU tetramer+ TCRβ+ cells remained slightly but consistently higher than the negative control stain with MR1-Ac-6-FP tetramer, suggesting the existence of a rare residual population of MR1-5-OP-RU-reactive T cells in *Traj33* KO mice. After depletion of immature CD24+ thymocytes, a population of MR1 tetramer+ T cells was clearly detected (Fig. 1a). These cells appeared to be approximately 50-fold less numerous than MAIT cells in WT mice. These cells also expressed CD44 suggesting that they were mature cells, and a similar population was not detected in *MR1* KO mice (Supplementary Fig. 4), suggesting that the residual MR1-5-OP-RU-reactive T cells in *Traj33* KO mice were MR1 dependent.

**MR1 tetramer+ T cells from *Traj33* KO mice.** To investigate whether the residual MR1 tetramer+ T cells could respond to 5-OP-RU, we devised a stimulation assay using plate-bound MR1-5-OP-RU monomers. Splenocytes from WT, *Traj33* het and *Traj33* KO mice were added onto MR1-5-OP-RU coated or uncoated in vitro culture plates for 5 days, and MR1-5-OP-RU tetramer+ cell expansion was measured via flow cytometry (Fig. 2). A clear population of MAIT cells expanded in these cultures, with a similar degree of expansion from the starting

population in each case (WT ~33-fold, het ~32-fold, KO ~35-fold). These cells did not stain with MR1 tetramer loaded with Ac-6-FP, confirming the 5-OP-RU Ag specificity of the expanded MR1 tetramer+ cells derived from *Traj33* KO mice (Fig. 2).

In order to determine which TCRs the *Traj33* KO-derived MR1-5-OP-RU tetramer+ cells were using, single cells were sorted from the in vitro-expanded population and their TCRs were sequenced by multiplex reverse transcription PCR[30] (Table 1 and Supplementary Table 1). In contrast to 5-OP-RU-expanded cells from WT spleen cultures, which were exclusively TRAV1/TRAJ33+, MR1-5-OP-RU tetramer+ cells from *Traj33* KO cultures all expressed TRAV1 rearranged with three alternate *Traj* genes: *Traj12* (22/40), *Traj9* (9/40), or *Traj40* (9/40). These data indicate that *Traj33* can be substituted by at least three other *Traj* genes in the context of the *Trav1* variable domain gene to form MR1-restricted, 5-OP-RU-reactive TCRs. These alternate *Traj* genes and the associated CDR3α regions showed patterns of conservation with the typical TRAV1/TRAJ33 MAIT TCR including a tyrosine at position 95 (Y95α) and a CDR3α region that was exactly 12 amino acids long. Thus, while these TCR-α chains lack TRAJ33, they still retained the integral CDR3α features known to be critical for MAIT TCR binding to MR1-5-OP-RU complex[16]. These cells are similar to the TRAV1-2/TRAJ12+ or TRAV1-2/TRAJ20+ MAIT cells that have been detected within the human MAIT TCR repertoire[10,11].

**Two classes of MR1-restricted T cell in *Traj33* KO mice.** That only three separate TCR-α chain sequences were detected in 5-OP-RU-expanded cultures of *Traj33* KO cells suggests only a limited array of alternative TRAJ genes that can substitute for TRAJ33 to form MR1-5-OP-RU-reactive TCRs. However, this limited diversity may reflect a bias associated with in vitro expansion in the presence of 5-OP-RU. Therefore, we tested whether MR1 tetramer+ TCRs could be directly identified ex vivo from *Traj33* KO mice. Paired productive TCRα and TCRβ sequences were obtained from 46 *Traj33* KO thymic MR1-5-OP-RU tetramer+ cells and 53 WT thymic MR1-5-OP-RU tetramer+ cells, respectively. A further 58 WT thymic MR1-5-OP-RU tetramer+ cells were sequenced for their TCRα usage only (Fig. 3a, b, Supplementary Tables 2, 3 and 4). From the *Traj33* KO cells, a range of TCR-α chains were detected that were broadly divided into two groups, consisting of TRAV1+ (20 cells) and TRAV1− (26 cells) sequences.

Of the 20 TRAV1+ cells, 11 were TRAV1/TRAJ9 and the remainder included TRAJ6 (1 cell), TRAJ12 (2 cells), TRAJ30 (3 cells) and TRAJ40 (3 cells). Of note, these all carried CDR3α regions that were 12 amino acids long and included a tyrosine at position 95 (Y95α) (Fig. 3, Supplementary Table 4). The 26 TRAV1− cells included TRAV3 (4 cells), TRAV6 (3 cells), TRAV9 (1 cell), TRAV15 (1 cell) and TRAV16 (17 cells). The major TRAV16+ population of cells were further divided into TRAJ18 (7 cells), TRAJ22 (2 cells), TRAJ26 (2 cells) and 1 each of TRAJ 52, 50, 40, 34, 17 and 4 (Supplementary Table 4). In contrast to the TRAV1+ group, the CDR3α loops of the TRAV1− TCRs were extremely variable in length (12–18 amino acids), and notably, only one of the CDR3α loops possessed the canonical tyrosine residue at position 95. Of the 111 WT cells we sequenced, all but 2 of these were TRAV1/TRAJ33+. The two remaining cells were both TRAV1/TRAJ9+, similar to the expanded cells observed post-antigenic stimulation (Fig. 2). The TCRβ chains for these MR1 tetramer+ T cells from *Traj33* KO were highly enriched (70%) for TRBV13 (Vβ8), and the remainder were a mix of TRBV19, 5, 4, 2 and 1. Of the 46 sequences derived from *Traj33* KO thymus samples, only 3

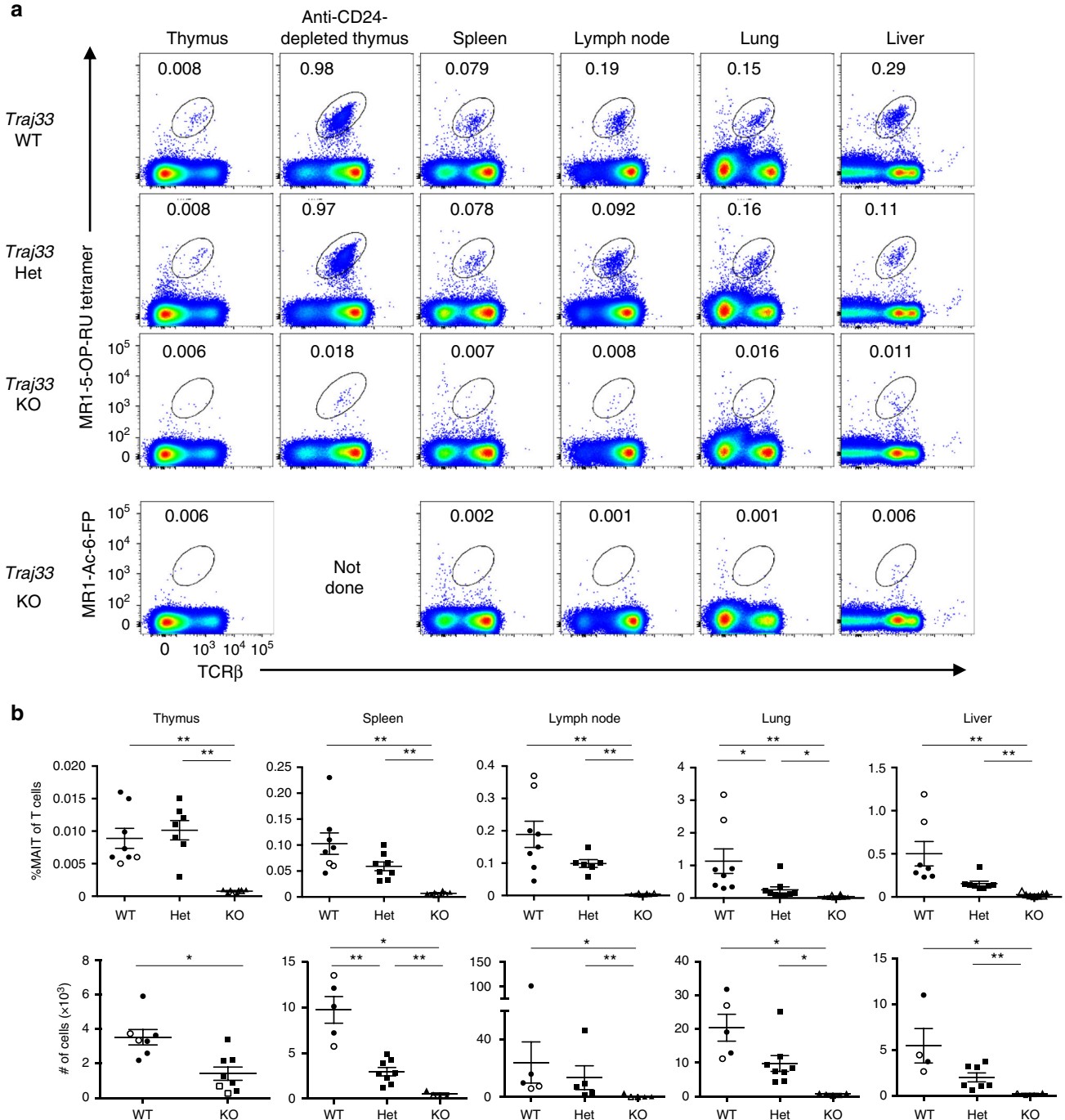

**Fig. 1** *Traj33* knockout (KO) mice have heavily reduced mucosal-associated invariant T (MAIT) cells **a** Flow cytometric analysis of MR1-5-OP-RU tetramer+ TCRβ+ MAIT cells from thymus and anti-CD24-complement-depleted enriched thymus, spleen, lymph node, lung and liver from *Traj33* wild-type (WT), *Traj33* heterozygous (het) and *Traj33* KO mice. Plots were pre-gated on B220− live lymphocytes, hence numbers show the percentage of gated MAIT cells of B220− cells. **b** Frequencies of all T cells and absolute numbers of MAIT cells from the aforementioned organs in the respective mice. Horizontal bars on scatter points signify mean ± SEM. Each scatter point represents an individual mouse, and data are derived from three independent experiments with a total of eight WT, eight *Traj33* het and eight *Traj33* KO mice. Source data are provided as a source data file. Open symbols represent ex-breeders, >20 weeks old. Statistical significance is based on *P ≤ 0.05, and **P ≤ 0.01 using Mann–Whitney rank-sum *U* test with a Bonferroni correction for three comparisons

were found to be repeats from identical clones, suggesting that intrathymic selection, rather than clonal expansion, gives rise to the majority of *Traj33* KO MR1 tetramer+ T cells. Taken together, these data demonstrate a higher degree of diversity within the MR1-restricted T cell population than is currently appreciated and, furthermore, that two broad groups of TRAJ33− MR1-restricted T cells exist: those that retain TRAV1 and a conserved CDR3α region including a tyrosine at position 95, and those that use diverse TRAV and TRAJ genes with highly variable CDR3α regions. Given the detection of TRAV1/TRAJ9 MAIT cells in WT mice, our data suggest that MAIT cells that express TRAJ33− TCRs are a normal part of the MR1-restricted repertoire but are largely outnumbered by TRAV1/TRAJ33+ cells within the MR1 tetramer+ population.

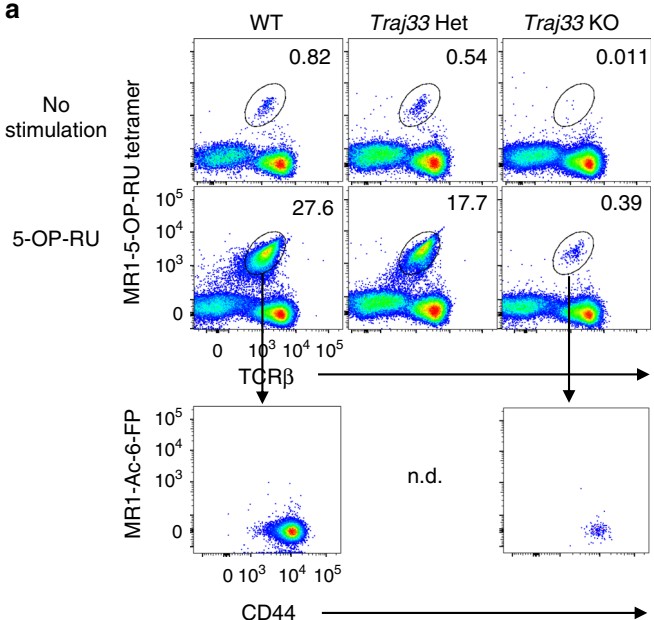

**Fig. 2** *Traj33* knockout (KO) MR1 tetramer+ cells expand following in vitro stimulation. Flow cytometric analysis of MR1-5-OP-RU tetramer+ TCRβ+ cells from *Traj33* wild-type (WT), het and KO splenocytes stimulated with plate-bound MR1-5-OP-RU monomers for 5 days. Data representative of three independent experiments, with a total of three WT, five *Traj33* het and five *Traj33* KO individual mouse spleen cultures. Plots were pre-gated on B220− live lymphocytes, hence numbers show the percentage of gated mucosal-associated invariant T cells of B220− cells. n.d. = not determined

**Table 1 Distribution of TRAV/TRAJ gene usage out of 16 cells sequenced from single-sorted MR1-5-OP-RU tetramer+ cells from WT splenic cultures and 40 cells from *Traj33* KO splenic cultures**

| Traj33 genotype | TRAV | TRAJ | Representative CDR3α | Frequency |
|---|---|---|---|---|
| WT | TRAV1 | TRAJ33 | CAVRDTNYQLIW | 16/16 |
| KO | TRAV1 | TRAJ12 | CAVRDGGYKVVF | 22/40 |
| KO | TRAV1 | TRAJ9 | CAVRDGGYKLTF | 9/40 |
| KO | TRAV1 | TRAJ40 | CAVQTGNYKYVF | 9/40 |

*KO knockout, WT wild type*

**TRAV1− MR1-reactive TCRs exhibit flexible Ag specificity.** In order to validate the MR1 reactivity of the atypical TCRs identified above and to determine their dependence on 5-OP-RU presented by MR1, we selected six TCRs to generate TCR-transfected HEK293T cell lines (Fig. 4). These lines included a classical TRAV1/TRAJ33 TRBV13-2 MAIT TCR from WT mice and from *Traj33* KO thymus: TRAV1/TRAJ9 TRBV13-3; TRAV1/TRAJ12 TRBV13-2; TRAV16/TRAJ18 TRBV13-2; TRAV6N-6/TRAJ31 TRBV13-1; TRAV3-4/TRAJ40 TRBV12-1. As a specificity control, a CD1d-restricted TCR TRAV13/TRAJ50 TRBV13-2[31] was also included. As expected, the classical MAIT TCR (TRAV1/TRAJ33) bound to MR1-5-OP-RU tetramer but not to MR1-Ac-6-FP tetramer or CD1d-α-GalCer tetramer. The other TRAV1+ cell lines with conserved CDR3 substitutions (TRAJ9 and TRAJ12) showed a very similar binding pattern to the classical MAIT TCR. In contrast, two of the non-conserved MR1-restricted TCRs: TRAV16/TRAJ18 and

TRAV6N-6/TRAJ31, bound not only to MR1-5-OP-RU tetramer but also to MR1-Ac-6-FP tetramer. One of the cell lines failed to bind to any of the tetramers, which may reflect the fact that the original cell from which this clone arose was CD8highCD44neg, in contrast to all the other clones from which TCRs were derived (based on index sorting analysis, Supplementary Fig. 5). As expected, the negative control NKT TCR did not bind to any of the MR1 tetramers but did bind to the CD1d-α-GalCer tetramer.

**MR1-reactive cells expand during infection in *Traj33* KO mice.** To directly test whether these atypical MR1-reactive T cells have the ability to respond to in vivo challenges, we carried out *Legionella* infection experiments as established in our recent paper[2]. Briefly, WT, *Traj33* KO and *MR1* KO mice were intranasally inoculated with 10^5 colony-forming units of *L. longbeachae* and the lungs were harvested 7 days post-infection (Fig. 5). MR1-5-OP-RU tetramer+ MAIT cells were markedly increased in the lungs of WT mice, and a clear expansion of atypical MR1 tetramer-reactive cells in the *Traj33* KO mouse lungs was also detected. No detectable population of MR1 tetramer+ cells was seen in *MR1* KO mouse lungs, suggesting that these responding cells in the *Traj33* KO mice were MR1 dependent (Fig. 5a, b). When probed for PLZF expression, these expanded cells in the *Traj33* KO lungs exhibited comparable levels of PLZF relative to WT lung MAIT cells (Fig. 5c). We next determined TCRα usage by sequencing 21 WT and 41 *Traj33* KO-expanded lung MR1-5-OP-RU tetramer+ cells (Table 2 and Supplementary Table 5). Twenty out of 21 of the WT sequences were TRAV1/TRAJ33+, and 1 atypical TRAV3/TRAJ35+ cell was detected. From the *Traj33* KO cells, 40 of the 41 cells expressed TRAV1 rearranged with one of three alternate TRAJ segments: TRAJ9 (19/41), TRAJ12 (9/41), or TRAJ30 (12/41). These all had 12 amino acid-long CDR3α regions incorporating Y95α (Table 2 and Supplementary Table 5), similar to those observed from in vitro-Ag-expanded population of *Traj33* KO cells (Fig. 2 and Supplementary Table 1). The single non-canonical TRAV1− sequence was TRAV7/TRAJ44, with a CDR3α length of 15 amino acids (Table 2 and Supplementary Table 5). These data demonstrate that atypical MR1-reactive cells that lack TRAJ33 can be activated and expand in response to microbial infection.

**Two classes of MR1-reactive TRAV1-2− T cells in humans.** The data above validate that the atypical MR1-reactive T cells detected in the *Traj33* KO mice can bind to MR1 via their TCRs. These fall into at least two groups, analogous to the human MR1-restricted T cell repertoire—one with conserved TRAJ substitutions and 5-OP-RU specificity, and another with diverse TRAV and TRAJ gene usage and potential for more diverse Ag reactivity. We next sought to probe the diversity of the human atypical MR1-restricted αβ T cell compartments from ex vivo human PBMC samples, with a focus on cell surface markers that align with a MAIT-like phenotype. This included three markers that are typically highly expressed on classical TRAV1-2+ MAIT cells[3]: the C-type lectin CD161[32], the IL-18Rα chain, CD218a[33], and the ectopeptidase CD26[5,34]. A cohort of 18 PBMC samples were stained with MR1-5-OP-RU tetramers and a panel of antibodies to identify TRAV1-2− αβ T cells with MAIT-like or non-MAIT-like surface markers (Fig. 6a). The MR1 tetramer staining pattern on TRAV1-2− MR1-5-OP-RU tetramer+ cells ranged in intensity from low to high (Fig. 6a; e.g. donors 2–3); however, some donors exhibited discrete populations of these cells (e.g. donor 1).

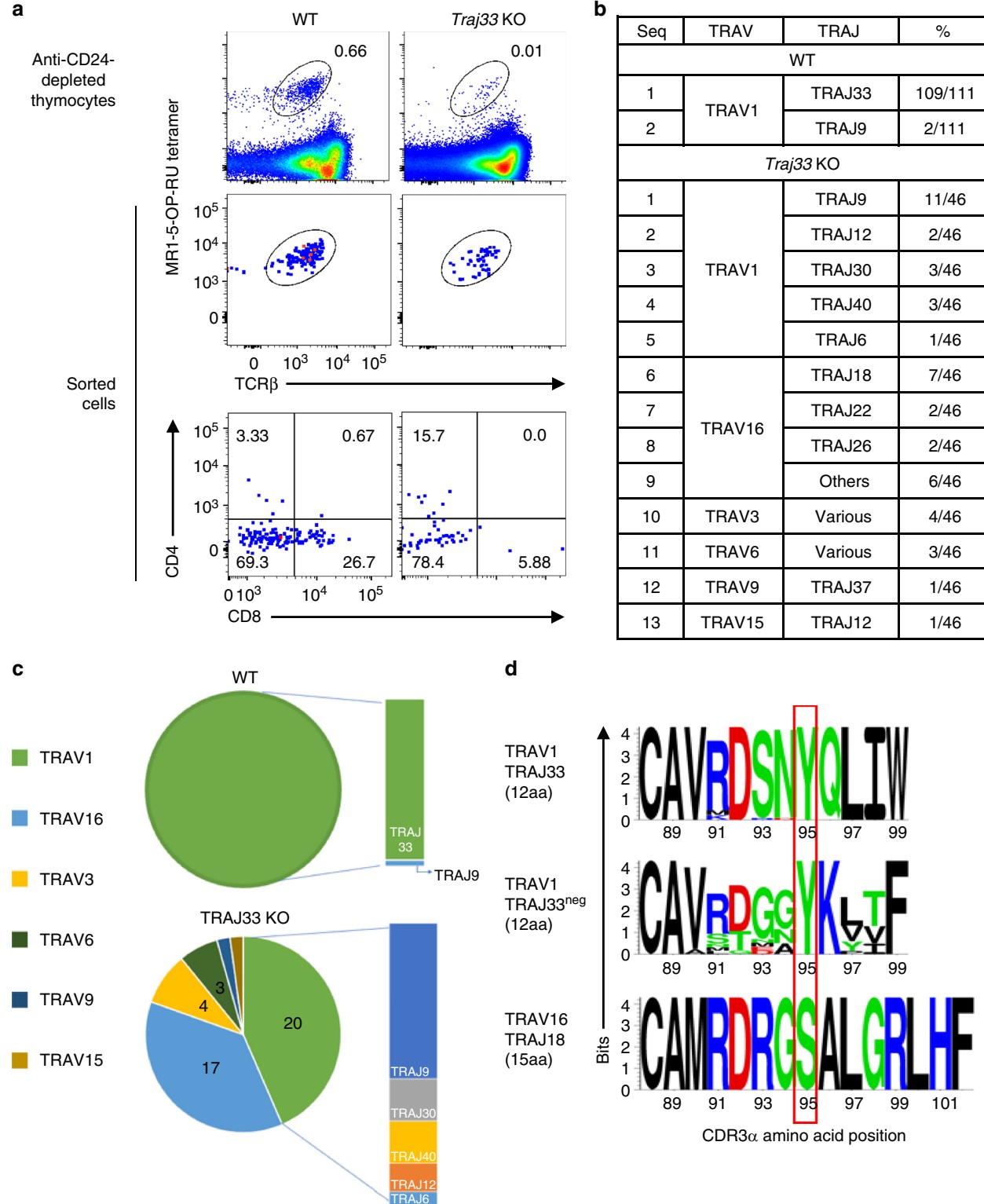

**Fig. 3** Analysis of residual MR1 tetramer+ cells from *Traj33* KO mice. **a** Flow cytometric plots of ex vivo enriched wild-type (WT) and knockout (KO) thymi. Data representative of a combined pool of 3–9 individual mouse thymus per strain. Top row plots were pre-gated on B220− live lymphocytes, second row plots depict combined index sorted single cells and third row plots show the CD4 and CD8 expression from these sorted cells. **b** Table lists the distribution of *Trav/Traj* genes out of 111 sequences obtained from sorted MR1-5-OP-RU tetramer+ WT cells and 46 sequences from *Traj33* KO cells. **c** Pie charts illustrate distribution of TRAV genes utilised from WT and KO mice, respectively. Adjacent bar charts show *Traj* gene usage by TRAV1+ MR1-5-OP-RU tetramer+ cells. **d** Sequence logos showing the amino acid distribution at the CDR3α junctions from T cell receptors (TCRs) analysed in **b**, including TRAV1/TRAJ33+ TCRs, TRAV1/TRAJ33− TCRs 12 amino acids in length and TRAV16/TRAJ18+ TCRs 15 amino acids in length. Red rectangle aligns amino acids at position 95 of the respective CDR3α sequences

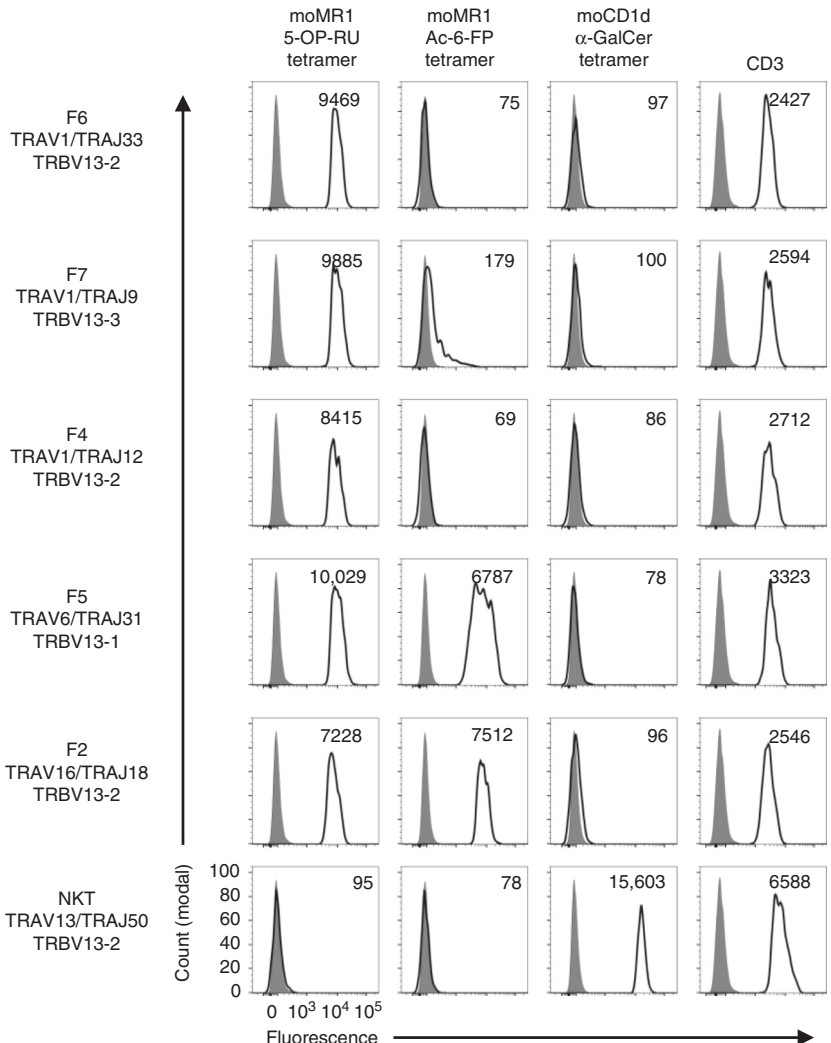

**Fig. 4** Validation of MR1 reactivity using T cell receptor (TCR) transfectants. Five paired TCRα- and β-chain transcripts from Fig. 3 and one irrelevant CD1d-α-GalCer-restricted TCR were selected for production of transiently transfected surface TCRs on HEK293T cells. Flow cytometric histogram overlays depict staining intensity of these respective transfected HEK293T cells stained with MR1-5-OP-RU, MR1-Ac-6-FP and CD1d-α-GalCer tetramers and are representative of two independent experiments. Cells were pre-gated for high levels of green fluorescent protein and surface TCR, and staining mean fluorescent intensity of gated population (transparent histograms) is depicted in top right of each histogram, overlaid on unstained cells (grey histograms)

As expected, the majority of TRAV1-2⁺ MAIT cells were CD218a$^{high}$ and CD161$^{high}$, in contrast to MR1-5-OP-RU tetramer⁻ 'conventional' αβ T cells, which were mostly low for these markers (Fig. 6a, second panels). For TRAV1-2⁻ MR1-5-OP-RU tetramer⁺ cells, two distinct populations emerged: some had high CD218a and CD161 expression, akin to MAIT cells (MAIT-like cells), and some had low expression of these markers (non-MAIT-like cells) (Fig. 6a, third panels). Further analysis showed that, similar to MAIT cells, MAIT-like TRAV1-2⁻ MR1-5-OP-RU tetramer⁺ cells also expressed high levels of CD26, whereas their non-MAIT-like counterparts were CD26$^{low}$.

We hypothesised that the similarities between MAIT-like TRAV1-2⁻ T cells and classical TRAV1-2⁺ MAIT cells may result from a common developmental pathway. A key feature of the MAIT cell developmental pathway is the expression of PLZF[35]. PLZF expression was measured by flow cytometry in 15 human blood samples, comparing classical TRAV1-2⁺ MAIT cells to MAIT-like and non-MAIT like MR1-restricted T cells (Fig. 6b). As expected, classical MAIT cells had higher median PLZF expression in comparison to conventional αβ T cells. MAIT-like TRAV1-2⁻ T cells expressed

levels of PLZF comparable to classical MAIT cells, whereas non-MAIT-like MR1-5-OP-RU tetramer⁺ T cells lacked PLZF expression (Fig. 6b).

In this cohort of donors, TRAV1-2⁺ MAIT cells were variable in frequency, accounting for a median of 2.4%, but ranging from 0.3% to 12.3% of total CD3⁺ αβ T cells (Fig. 6c). The TRAV1-2⁻ MR1-5-OP-RU tetramer⁺ cells were less frequent (median of 0.02%) and ranging from 0.01% to 0.12% of total CD3⁺ αβ T cells (Fig. 6c). Of the TRAV1-2⁻ cells, MAIT-like cells were typically less frequent than non-MAIT-like cells, but both were as high as 0.08% and 0.06% of total CD3⁺ αβ T cells, respectively (Fig. 6c). As a proportion of total MR1-5-OP-RU tetramer⁺ αβ T cells, TRAV1-2⁺ MAIT cells were by far the most abundant, accounting for a median of 98.7% of cells, whereas TRAV1-2⁻ cells accounted for a median of 0.78%, which was made up by MAIT-like and non-MAIT-like cells with median frequencies of 0.12% and 0.62%, respectively (Fig. 6c).

**Co-receptor usage by TRAV1-2⁻ MR1-reactive T cell subsets.** The majority of TRAV1-2⁺ MAIT cells are CD8⁺ or CD4⁻CD8⁻

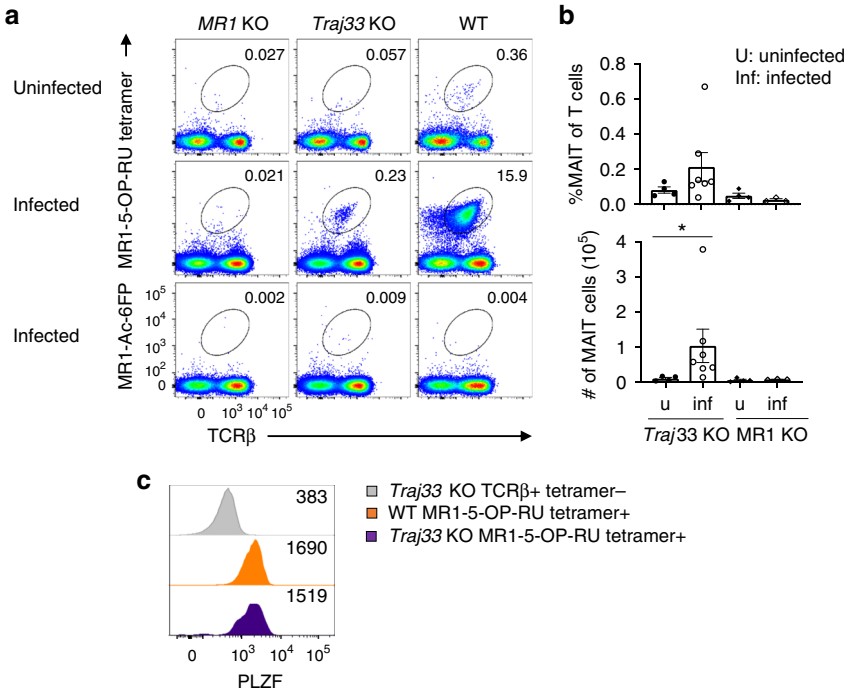

**Fig. 5** *Traj33* knockout (KO) MR1 tetramer⁺ cells respond to *L. longbaechae* in vivo. **a** Flow cytometric analysis of MR1-5-OP-RU tetramer⁺ or MR1-Ac-6-FP tetramer⁺ TCRβ⁺ cells from the lungs of *MR1* KO, *Traj33* KO and wild-type (WT) mice either uninfected or infected with $10^5$ CFU of *L. longbaechae* for 7 days. Plots were pre-gated on B220⁻ live lymphocytes, hence numbers show the percentage of gated mucosal-associated invariant T (MAIT) cells of B220⁻ cells. **b** Frequencies of all T cells and absolute numbers of MAIT cells from the aforementioned lungs in the respective mice. Horizontal bars on scatter points signify mean ± SEM. Each scatter point represents an individual mouse, and data are representative of two independent experiments, with a total of four uninfected *Traj33* KO, seven infected *Traj33* KO, four uninfected *MR1* KO and three infected *MR1* KO individual mouse lungs. Source data are provided as a source data file. Statistical significance is based on *$P \leq 0.05$ using Mann–Whitney rank-sum $U$ test between naive and infected samples. **c** Flow cytometric histogram overlay showing PLZF staining on MR1-5-OP-RU tetramer⁺ TCRβ⁺ cells from infected WT and *Traj33* KO lung, compared to tetramer⁻ TCRβ⁺ cells and are representative of three mice per group

**Table 2 The distribution of TRAV-TRAJ gene usages out of 21 cells sequenced from single-sorted MR1-5-OP-RU tetramer⁺ cells from infected WT lung and 41 cells from infected *Traj33* KO lungs**

| Traj33 genotype | TRAV | TRAJ | Frequency |
|---|---|---|---|
| WT | TRAV1 | TRAJ33 | 20/21 |
| WT | TRAV3 | TRAJ35 | 1/21 |
| KO | TRAV1 | TRAJ9 | 19/41 |
| KO | TRAV1 | TRAJ12 | 9/41 |
| KO | TRAV1 | TRAJ30 | 12/41 |
| KO | TRAV7 | TRAJ24 | 1/41 |

*KO* knockout, *WT* wild type

double negative (DN), although minor populations of CD4⁺ single positive (SP) or CD4⁺CD8⁺ DP MAIT cells are also present in human blood[3]. Here we determined whether MAIT-like and non-MAIT-like TRAV1-2⁻ MR1-5-OP-RU tetramer⁺ cells exhibit differential co-receptor distribution (Fig. 6d). As expected, the majority of MAIT cells were CD8⁺ (median 74.4%) or CD4⁻CD8⁻ DN (median 22.5%). MAIT-like TRAV1-2⁻ cells were generally similar to MAIT cells, with the majority of cells being CD8⁺ SP (median 64.5%) and most of the remaining cells being DN (median 29.1%), with few CD4⁺ SP or CD4⁺CD8⁺ DP cells. However, in comparison to classical MAIT cells, CD8 expression was more variable between donors, with the

proportion of CD8⁺ cells ranging from 3% to 100% and from 47% to 89% for MAIT-like and MAIT cells, respectively. Non-MAIT-like cells were more enriched for CD8 expression, with a median of 81.7% and range of 30.8–92.9. In contrast to MAIT and MAIT-like cells, the remaining CD8⁻ non-MAIT-like cells were distributed between CD4⁺ SP, CD4⁺CD8⁺ DP and CD4⁻CD8⁻ DN subsets, with medians of 9.3%, 3.0% and 2.6%, respectively. In summary, in the TRAV1-2⁻ MR1-5-OP-RU⁺ compartment, MAIT-like cells are similar to MAIT cells, whereas non-MAIT-like cells are more highly enriched for CD8⁺ cells, then CD4⁺ cells, with no enrichment for DN T cells.

**Ag-specificity of human TRAV1-2⁻ MR1-restricted T cells.** To test the Ag reactivity of MAIT-like TRAV1-2⁻ MR1-restricted cells, these samples were stained with MR1-5-OP-RU or MR1-6-FP tetramers, as well as CD218a and CD161 to distinguish between MAIT-like and non-MAIT-like cells (Fig. 6e). In 10/15 donors, MAIT-like TRAV1-2⁻ cells were only stained by MR1-5-OP-RU tetramers (e.g. donor 4, Fig. 6e) while in the remaining 5 donors, some of these cells could also be labelled by MR1-6-FP tetramers (e.g. donor 2, Fig. 6e). Notably, most of the MR1 6-FP tetramer⁺ TRAV1-2⁻ T cells fell into the non-MAIT-like category, lacking both CD161 and CD218a.

**A canonical, public, invariant TRAV1-2⁻ human MAIT TCR.** To gain insight into the TCR repertoire of TRAV1-2⁻ MR1-restricted T cells and to compare the repertoire between MAIT-like (CD161⁺, CD218a⁺, CD26⁺) and non-MAIT-like subsets of

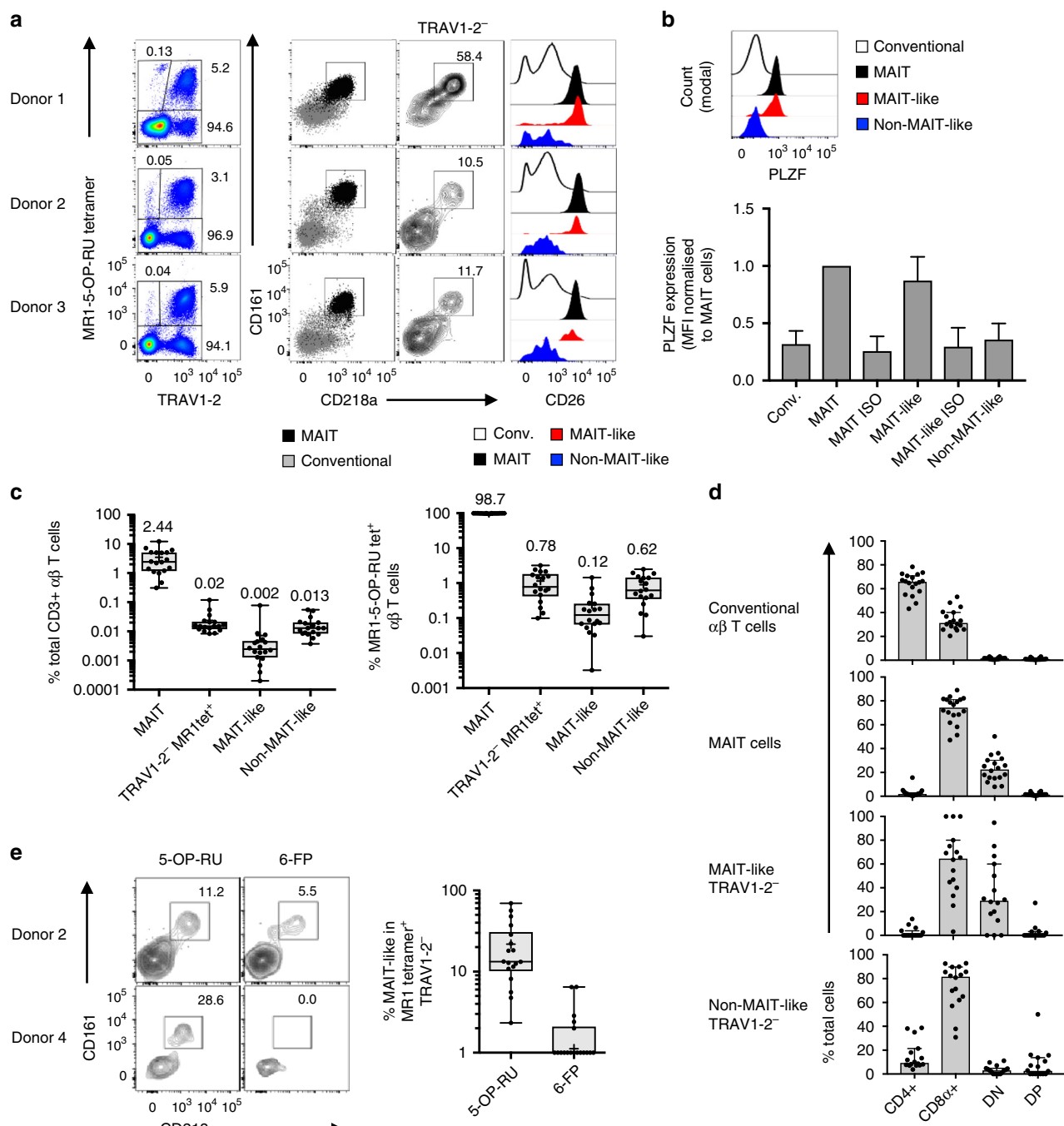

**Fig. 6** TRAV1-2⁻ MR1 tetramer⁺ αβ T cells in humans. **a** Flow cytometric only the first column is colour plots from three representative peripheral blood mononuclear cell (PBMC) samples showing gating strategy of mucosal-associated invariant T (MAIT)-like and non-MAIT-like T cells. In the second column, MAIT cells (black) are overlaid above conventional T cells (grey). In the third column, TRAV1-2⁻ MR1 tetramer⁺ cells are shown. In the fourth column, histogram overlays show populations as depicted in the key below. **b** Flow cytometric histogram overlay showing PLZF staining on T cell subsets from one representative PBMC sample. Bar graph showing PLZF expression (normalised to MAIT cells) from a cohort of PBMC samples. **c** Box and whisker plots showing the proportion of MAIT cells, total TRAV1-2⁻ MR1-5-OP-RU tetramer⁺ cells, MAIT-like TRAV1-2⁻ MR1-5-OP-RU tetramer⁺ cells and non-MAIT-like TRAV1-2⁻ MR1-5-OP-RU tetramer⁺ cells as a percentage of total CD3⁺ αβ T cells (left), and a percentage of MR1-5-OP-RU tetramer⁺ αβ T cells (right) with median values depicted above boxes. **d** Bar graphs showing the co-receptor distribution on T cell subsets from PBMC samples. **e** Flow cytometric contour plots showing CD218a and CD161 expression on TRAV1-2⁻ MR1-5-OP-RU tetramer⁺ (left panel) and MR1-6-FP tetramer⁺ cells (right panel) from two representative donors. Box and whisker plots showing the proportion of MAIT-like cells (CD218a⁺ CD161⁺) in TRAV1-2⁻ MR1-5-OP-RU tetramer⁺ and MR1-6-FP tetramer⁺ cells (data points on axis = 0). In **a**, **e**, n = 18 and in **b**–**d**, n = 15 donors analysed across 3 independent experiments each. For box and whisker plots, boxes show median, upper and lower quartiles, whiskers represent min. and max. points and '+' represents mean. For bar graphs, error bars represent median ± 95% CI. Source data are provided as a Source Data file

**Table 3 List of TCR sequences from human TRAV1-2− MR1-5-OP-RU tetramer+ αβ T cells**

| | TRAV | CDR3α | TRAJ | TRBV | CDR3β | TRBJ | Freq | Donor |
|---|---|---|---|---|---|---|---|---|
| MAIT-like | 5 | C**SRGSS**NTGKLIF | 37 | 19 | CAS**PKARF**YEQYF | 2-7 | 1/27 | 2 |
| TRAV36− | 8-6 | CAVS**DR**GGSNYKLTF | 53 | 6-4 | CASSD**GGEG**YNEQFF | 2-1 | 4/27 | 2 |
| | 9-2 | CALS**DHE**AAGNKLTF | 17 | 24-1 | CATS**EAGD**NEQFF | 2-1 | 1/27 | 2 |
| | 12-2 | CA**FAS**GYSSASKIIF | 3 | 4-1 | CASSPDFDPRDTQYF | 2-3 | 1/19 | 3 |
| | 13-2 | CAE**KLG**GGGNKLTF | 10 | 12-4 | CAS**RPGQGW**ETQYF | 2-5 | 1/18 | 4 |
| | 27 | CA**VS**NAGKSTF | 27 | 6-6 | CA**ISQVSSSYK**EQFF | 2-1 | 1/19 | 3 |
| MAIT-like | 36 | CA**VY**NTDKLIF | 34 | 28 | CASSL**WAIQ**ETQYF | 2-5 | 2/27 | 2 |
| TRAV36+ | 36 | CA**VY**NTDKLIF | 34 | 28 | CASSL**YYS**QETQYF | 2-5 | 11/18 | 4 |
| | 36 | CA**A**YNTDKLIF | 34 | 28 | CASS**PSSF**QETQYF | 2-5 | 2/27 | 2 |
| | 36 | CA**T**YNTDKLIF | 34 | 28 | CASSL**FGR**QETQYF | 2-5 | 11/19 | 3 |
| | 36 | C**VP**YNTDKLIF | 34 | 28 | CASS**PWDY**QETQYF | 2-5 | 1/27 | 2 |
| | 36 | CA**VY**NTDKLIF | 34 | 25-1 | CAS**SSLWGV**ETQYF | 2-5 | 2/27 | 2 |
| | 36 | CA**PY**NTGKLIF | 37 | 28 | CASSL**LAS**QETQYF | 2-5 | 21/29 | 1 |
| | 36 | CA**GY**NTGKLIF | 37 | 28 | CAS**SQWTE**QETQYF | 2-5 | 6/29 | 1 |
| Non | 4 | CL**GGLT**SNYQLIW | 33 | 24-1 | CATS**APGLA**YNEQFF | 2-1 | 1/18 | 3 |
| MAIT-like | 9-2 | CA**LRV**NNAGNMLTF | 39 | 20-1 | CSARE**SGKD**TEAFF | 1-1 | 3/18 | 3 |
| | 12-3 | CAMS**VE**DYKLSF | 20 | 11-3 | CASSL**DQGAD**TGELFF | 2-2 | 1/18 | 3 |
| | 13-2 | CAE**GA**SGYSTLTF | 11 | 24-1 | CATSDL**YDG**EAFF | 1-1 | 1/17 | 2 |
| | 14 | CAMRE**Y**SGYALNF | 41 | 28 | CASSL**WQ**STDTQYF | 2-3 | 2/17 | 2 |
| | 19 | CALSE**EVS**AGGTSYGKLTF | 52 | 4 | CASSQ**DWGPPG**NEQFF | 2-1 | 1/17 | 2 |
| | 19 | CA**LRVKV**GGSYILTF | 6 | 4-1 | CASSQ**DRAGGRF**SGNTIYF | 1-3 | 1/18 | 3 |
| | 20 | CA**G**SGGSYIPTF | 6 | 3 | CASS**SH**NEQFF | 2-1 | 2/18 | 3 |
| | 36 | C**GVDG**ARLML | 31 | 29-1 | CSV**MTGF**TEAFF | 1-1 | 2/18 | 3 |

Bold text = non-germline-encoded amino acids
*TCR* T cell receptor

these cells, single-cell TCR sequencing was performed on these cells from four unrelated blood donors (Table 3). The non-MAIT-like cells used highly diverse TRAV, TRAJ, TRBV and TRBJ genes to encode variable TCR-α and -β chains. Moreover, there was no conservation in CDR3α or CDR3β junctional motifs or lengths. Interestingly, the MAIT-like cells distributed into two subsets in terms of TCR usage, one with diverse TCR gene usage, CDR3 junctional motifs and length. The remaining MAIT-like TCRs were exclusively TRAV36+, 6/8 used TRAJ34, while the remaining 2 used TRAJ37. Furthermore, 7/8 of these TCRs used TRBV28 while one used TRBV25-1. All eight of these TRAV36+ TCRs used TRBJ2-5. Both the CDR3α and CDR3β had invariant lengths of 11 and 14 amino acids, respectively, with highly germline-encoded CDR3α and semi-invariant CDR3β sequence motifs (Fig. 7a, b). Moreover, this canonical pairing was observed in 4/4 donors and was identical to that of a clone (MAV36) that we had previously characterised among in vitro-expanded cells from a different donor[24]. Indeed, in 4/8 donors, these TCRs were highly clonally expanded and represented as many as 21/29 TCRs sequenced from one donor. Accordingly, the TRAV1-2− MAIT-like population is enriched for a public, canonical, invariant TRAV36+ TCR.

Closer analysis of these TRAV36+ TCRs (including the previously identified MAV36 TCR) revealed that the TCR-α chain can be formed from fully germline-encoded DNA, with only two amino acids at the TCR-α V-J gene junction varying. At position 90, most TCRs encoded an alanine; however, a valine substitution was present in one TCR, and at position 91, amino acids with short side chains were permitted, including valine, proline, alanine, threonine and glycine (Fig. 7a, b). Furthermore, the two TCRs that utilised TRAJ37 rather than TRAJ34 incorporated non-germline encoded *n* nucleotides at the CDR3α junction such that a tyrosine was formed at position 92—a residue that is germline encoded in TRAJ34. This resulted in a conserved amino acid motif of CXXYNTXKLIF. In our previous structural analysis of a TRAV36/TRAJ34+ MR1-5-OP-RU ternary complex[24], the residue (aspartic acid) at position 95

(D95α) did not play a role in docking, whereas Y92α, N93α and T94α were involved in the network of molecular interactions at the TCR–MR1-Ag interface. Thus incorporation of glycine at position 95α in the TRAJ37+ TCRs is unlikely to impact on MR1 reactivity, while the Y92α, N93α and T94α residues are fixed. Notably, no other human TRAJ genes encode this sequence, providing a possible basis for TRAJ34/37 gene usage. The TCR-β chain was also invariant in length with amino acid variability detected at positions 95–99β, whereas TRBJ2-5 gene-encoded E100–F104β were also invariant. This is also consistent with our previous structural data using the MAV36 TCR[24], where E100β and T101β played a direct role in docking, whereas amino acids in positions 95–99β were not extensively involved, thereby providing a possible explanation for diverse sequence usage at positions 95–99β but conserved CDR3β length and TRBJ2-5 gene usage.

**Specificity of TRAV36 MAIT TCRs.** To validate the MR1-Ag restriction of the TRAV36+ TCRs, three of them were cloned into pMIGII expression vectors, including MAV36-1 (TRAV36/ TRAJ34 TRBV25-1), MAV36-2 (TRAV36/TRAJ34 TRBV28) and MAV36-3 (TRAV36/TRAJ37 TRBV28). These vectors were then used to co-transfect HEK293T cells along with a pMIGII vector encoding the human CD3 subunits to induce surface expression of the new TCRs. Transfected cells were then stained with a panel of tetramers to determine the MR1-Ag specificity of the clones. As controls, a TRAV1-2+ MAIT TCR using the TRBV28 gene (clone MBV28[24]), an MR1-6-FP-reactive TRAV1-2+ MAIT TCR (clone M33-64[24]), the original MAV36 TCR[24] and a type I NKT TCR as a negative control (clone NKT15[36]) were also included (Fig. 7c). As expected, all transfectants showed a range of staining for GFP/CD3, so a gate was placed on the highest TCR-expressing transfectants in order to compare each cell line where TCR levels were comparable (Supplementary Fig. 6). The specificity control type I NKT TCR stained with huCD1d-α-GalCer tetramers but failed to bind any of the MR1 tetramers. Conversely, the TRAV1-2+ MAIT TCRs and the MAV36 TCR bound human MR1-5-OP-

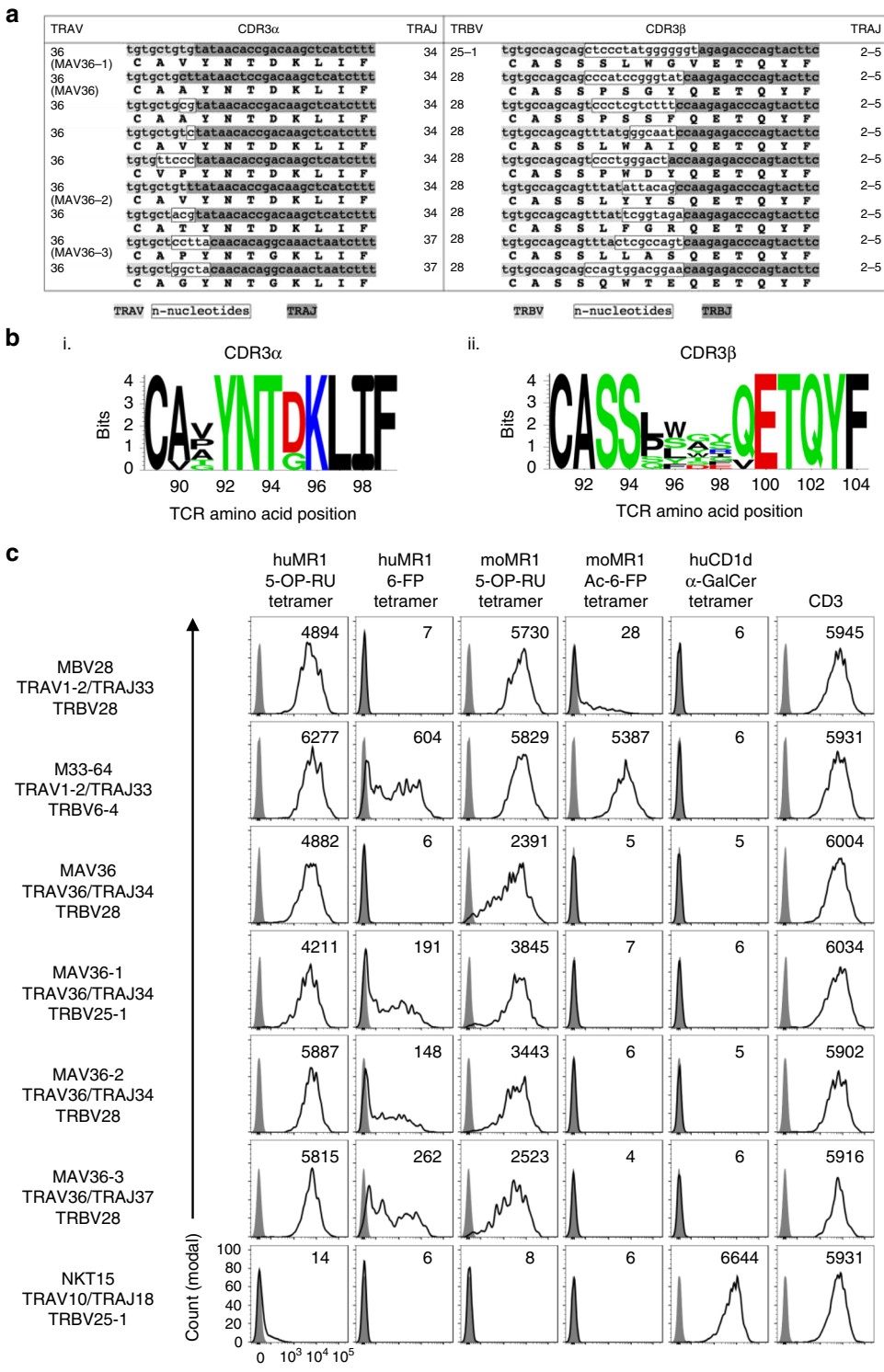

**Fig. 7** MR1-restricted TRAV36+ T cell receptors (TCRs) express canonical TCR-α and TCR-β chains. **a** Table showing CDR3α and CDR3β nucleotide and amino acid sequences from nine TRAV36+ TCRs. Clone names of clones further characterised in this study are written underneath TRAV gene. Germline-encoded residues are high in grey (variable genes = light grey; joining genes = dark grey) and n nucleotides are highlighted in a white box. **b** Sequence logos showing the amino acid distribution at the CDR3α and CDR3β junction from n = 9 TRAV36+ TCRs. **c** Flow cytometric histogram overlays showing MR1 (columns 1–4) tetramers, CD1d (column 5) tetramer and CD3 (column 6) staining on HEK293T cells transiently transfected to express human TCRs (left column). Stained cells (transparent histograms) are overlaid on unstained cells (grey histograms). Stained cells are gated for high levels of green fluorescent protein, and mean fluorescent intensity of gated population is depicted in top right of each histogram. Data are representative of two independent experiments

RU tetramers but failed to bind CD1d-α-GalCer. The TRAV36 TCRs were largely, but not absolutely, Ag dependent, binding strongly to human MR1-5-OP-RU tetramers with similar intensity to the classical TRAV1-2/TRAJ33 MAIT TCRs and moderately to human MR1-6-FP tetramers, similar to the previously described TRAV1-2/TRAJ33 M33-64 TCR[24]. Interestingly, the MAV36 TCR that we had previously defined[24] did not bind to human MR1-6-FP tetramer, suggesting that the subtle differences in CDR3 amino acid sequence may modulate Ag responsiveness. Each of the TRAV36 TCRs exhibited xenoreactivity with mouse MR1 tetramers loaded with 5-OP-RU but not when loaded with Ac-6-FP, in contrast to the M33-64 TCR. Accordingly, we have identified a population of MR1-5-OP-RU-restricted, TRAV1-2⁻ αβ T cells that utilise a novel, canonical, public, TRAV36⁺ TCR and exhibit phenotypic features akin to MAIT cells.

## Discussion

Because mouse MAIT cells are all thought to express an invariant TRAV1/TRAJ33 TCR-α chain, we generated *Traj33* KO mice in order to prevent the development of these cells and to generate a new mouse model for the study of these cells in health and disease. As expected, we found that MAIT cells are markedly diminished in *Traj33* KO mice. However, a residual population of mature MR1 tetramer-reactive T cells was detected in these mice. Through TCR sequencing studies of in vitro expanded cells, direct ex vivo analysis of residual MR1-restricted T cells and examination of in vivo microbial responsive cells, we have determined the presence of two groups of MR1-reactive T cells in *Traj33* KO mice: (i) TRAV1⁺ cells that used a small number of alternate TRAJ genes (TRAJ6, TRAJ9, TRAJ12, TRA30 and TRAJ40), allowing the formation of the conserved CDR3α loop comprising exactly 12 amino acids and tyrosine at position 95 (Y95α), and (ii) TRAV1⁻ cells that used a broad range of TRAV and TRAJ genes, with variable CDR3α length and no tyrosine at position 95. The former population appears to be similar to human TRAV1/TRAJ12/TRAJ20 MAIT cells, which represent a subset of classical MAIT cells in human blood with conserved TRAJ CDR3α length and Tyr at position 95[7,10,11,21]. These alternate TRAJ genes are not known to alter the specificity of MAIT cells in humans nor do they appear to have had any major impact on the specificity of the mouse TRAJ33⁻ MAIT cells. However, they can pair with different CDR3β, thereby potentially indirectly affecting Ag specificity[3]. The latter population demonstrates that diverse TCR-α chains expressing various TRAV and TRAJ genes can support MR1 reactivity, and furthermore, this can result in variable ability to detect MR1-bound Ags, as evidenced by the ability of some of these to bind to MR1 tetramer loaded with Ac-6-FP. Consistent with classical mouse MAIT cells, both types of residual mouse MR1 tetramer⁺ cells were heavily biased towards TRBV13 usage. This also implicates a role of the TCR-β chain in influencing the development of MR1-reactive TCRs regardless of their TCR-α chain composition.

The reason why TRAJ33⁻ MAIT cells have not been previously detected in mice is probably because they are very infrequent, even compared to TRAJ33⁺ MAIT cells which themselves are quite rare in mice[12]. Indeed, after sequencing 111 WT thymic MAIT cell clones ex vivo, we found two that expressed TRAJ9, and another atypical TCR-α sequence was detected in the in vivo-expanded MAIT cells from WT mice following *Legionella* infection. These findings strongly suggest that alternative MR1-restricted TCRs exist, albeit infrequently, in normal mice. Our data raise the important question of whether MAIT cells occupy a specific niche in mice. If this was the case, we might have expected the TRAJ33⁻ MR1-reactive cells in *Traj33* KO mice to have expanded to similar frequencies as MAIT cells in WT mice,

but this was clearly not the case. Indeed, even in *Traj33* het mice, we saw a trend towards fewer MAIT cells that was statistically significant in the spleen and lung, suggesting that there is little pressure for MAIT cells to occupy a specific niche. Furthermore, while the TCR-β chain was heavily biased towards TRBV13 usage in the MR1 tetramer⁺ cells in *Traj33* KO mice, there was no evidence of in vivo clonal expansion as indicated by diverse TRBJ and CDR3β usage.

While the scarcity of these cells made it difficult to undertake a detailed phenotypic analysis of the residual MR1-restricted T cells in *Traj33* KO mice, we were able to determine that they were CD24⁻CD44⁺, suggesting that they had undergone intrathymic maturation to resemble stage 3 MAIT cells (CD24⁻CD44⁺), where the acquisition of PLZF instils the expression of CD44 and effector function[35,37]. These cells also displayed a similar CD4/CD8 co-receptor profile to stage 3 MAIT cells. Therefore, these residual MR1-restricted T cells may undergo a parallel differentiation pathway to that followed by classical mouse MAIT cells, thereby representing a mouse equivalent to non-classical MAIT cells found in humans[7]. Furthermore, we show that these residual cells in the lung were able to respond to pulmonary *Legionella* challenge and expand as a PLZF⁺ population, suggesting that, while very infrequent in unchallenged clean mice, these cells are capable of responding and expanding in vivo if given an appropriate antigenic stimulus[2]. As no equivalent population was detected in *Legionella* challenged *MR1* KO mice, this supports the concept that the MR1 tetramer⁺ cells in the *Traj33* KO mice are MR1 restricted.

We and others have previously described MR1 tetramer⁺ TRAV1-2⁻ T cells in humans[24–26]. Here we have further probed the human TRAV1-2⁻ MR1-reactive T cell compartment directly ex vivo, which revealed the existence of two broad populations of these cells. One population ('MAIT-like' cells) phenotypically resembled MAIT cells with high expression of CD161, CD218a and CD26; a predominantly CD8⁺ and DN co-receptor profile and expression of the transcription factor PLZF. The other population of 'non-MAIT-like' cells lacked CD161, CD218a and CD26 and did not express PLZF. These cells were predominantly CD8⁺ although some were also CD4⁺ and few were DN. Non-MAIT-like cells were readily detected even when using MR1-6-FP tetramers, while the MAIT-like cells were only detected in a subset of donors with MR1-6-FP tetramers. Accordingly, MAIT-like cells fit into our recently proposed classification system as non-classical MAIT cells, whereas the non-MAIT-like cells align with a classification as atypical MR1-restricted T cells[7]. It is likely that the family of TRAV1-2⁻ MR1-reactive cells encompasses altered and potentially broader specificity for other microbial and/or non-microbial ligands in association with MR1, as proposed in previous studies[7,19,24,26]. Furthermore, the TRAV1-2⁻ MR1-reactive T cells that do not express classical MAIT cell molecules (CD161, CD218 and PLZF) may have distinct developmental origins based on their specificity during intrathymic selection, potentially instilling these cells with a phenotype and function more aligned with conventional T cells. In this study, we explored MR1-reactive TRAV1-2⁻ T cells using single-cell TCR sequencing, which has allowed us to examine paired TCR-α and β chains and also to produce and study cell lines expressing these TCRs. A limitation of this approach is the depth of sequencing. TCR deep sequencing has previously been applied to MR1-reactive cells, revealing further diversity than is commonly appreciated in the TCR usage by these cells although it is difficult to validate the specificity of these unpaired TCR chains[3,11]. Nonetheless, further studies like these will be very valuable in gaining a thorough understanding of the scope of the MR1-reactive T cell repertoire and the extent of the different Ags that can be seen by these cells.

While both the MAIT-like and non-MAIT-like populations in humans carried a diverse TCR-repertoire, the MAIT-like subset also included cells with an invariant TRAV36 TRAJ34/TRAJ37+ TRBV28/TRBJ2-5+ subset, almost identical to a clonally expanded population of TRAV36/TRAJ34+ MR1-5-OP-RU-reactive cells we had previously identified from a single donor[24]. Here we have detected these cells in four unrelated human donors, indicating a public TCR repertoire. The extremely high TCR conservation involving the TRAV, TRAJ, TRBV and TRBJ genes, along with CDR3α and CDR3β amino acid motifs and lengths, represents the first description of an αβ T cell population with canonical TCRα and TCRβ chain usage. This suggests major molecular constraints dictating MR1-5-OP-RU recognition by these TCRs, with little room for variation relative to the classical TRAV1-2+ MAIT TCR repertoire, which may be why non-classical TRAV36+ MAIT TCRs are rare in comparison to classical TRAV1-2+ MAIT TCRs. Nonetheless, these TCRs utilise a different docking strategy to the TCRs of classical MAIT cells and thus may also permit recognition of distinct Ags beyond 5-OP-RU. This concept is also reminiscent of how variations in TCR-α and TCR-β chain usage within CD1d-α-GalCer-reactive NKT cells can differentially impact on the hierarchy of other lipid Ags detected by these cells[31,38–40].

Taken together, our data demonstrate that there are multiple TCR-α chain conformations in mice and humans that can imbue MR1 reactivity upon developing T cells in the thymus. This TCR diversity gives rise to two broad populations of cells, some that resemble MAIT cells and some that are markedly distinct from MAIT cells. The range of Ag specificities and functional potential of these TRAV1-2− MR1-restricted T cells represents an important area for future studies. Indeed, as we learn more about distinct Ags that can be presented by MR1, we may discover other populations of MR1-restricted T cells that we are missing with the use of MR1-5-OP-RU and MR1-6-FP tetramers. Given the spacious Ag-binding groove of MR1 that is capable of accommodating Ags much larger in size than 5-OP-RU and 6-FP, studies into the full scope of MR1-restricted Ags and the corresponding MR1-restricted TCR repertoire will be important to properly understand this arm of the immune system.

## Methods

**Mice**. The *Traj33* gene was deleted in C57BL/6 blastocysts via CRISPR/Cas9 deletion guided by flanking single-guide RNA motifs[41], as shown in Supplementary Fig. 1A. *Traj33* chimeric founder mice were generated at the Walter and Eliza Hall Institute (WEHI) Animal Facility and imported into the Department of Microbiology and Immunology Biological Resource Facility, University of Melbourne at the Peter Doherty Institute for Infection and Immunity. Chimeric founder mice were backcrossed for $n = 1$ generation onto C57BL/6 WT mice and subsequently intercrossed to obtain *Traj33* Het and *Traj33* KO mice. All animal experimentation was approved by the University of Melbourne Animal Ethics Committee or the WEHI Animal Ethics Committee.

**Human samples**. Human buffy coats from healthy blood donors were obtained, with written informed consent, from the Australian Red Cross Blood Service after approval from the University of Melbourne Human Ethics Committee (1035100). Buffy coats were processed by standard density gradient using Ficoll-paque Plus (GE Healthcare) and cryopreserved in liquid nitrogen for subsequent use.

**Organ preparation and cell suspensions**. Single-cell suspensions of thymus, spleen and lymph nodes were prepared by mechanically dissociating each individual organ through a 30-μm nylon mesh MACS SmartStrainer (Miltenyi Biotec) into cold fluorescence-activated cell sorting (FACS) buffer (phosphate-buffered saline (PBS) with 2% foetal calf serum (FCS)). Splenocytes were subjected to red blood cell lysis before resuspension into FACS buffer. Lung and liver tissues were perfused with 10 ml PBS immediately after mice were sacrificed. Lungs were repeatedly sheared into small pieces prior to enzymatic digestion with 3 mg/ml collagenase type III (Worthington Biochemical Corporation) supplemented with 2% FCS, at 37 °C for 60 min. Perfused livers were mechanically dissociated through 70-μm nylon mesh MACS SmartStrainers and then purified for lymphocytes with a 33% isotonic Percoll (GE Healthcare) gradient.

**Anti-CD24-mediated depletion of immature thymocytes**. Thymus suspensions were incubated with anti-CD24 (clone J11D, also known as heat-stable Ag, produced in-house from J11D hybridoma) at 4 °C for 30 min. This was followed by a 30-min incubation at 37 °C with Rabbit Complement (GTI Diagnostics Wisconsin) and 1 mg/ml DNAse (Roche) to deplete anti-CD24-bound immature thymocytes. Viable thymocytes were then purified on a Histopaque-1083 (Sigma) density gradient and resuspended in FACS buffer.

**Staining of cells for flow cytometry**. PBMC samples were incubated with human Fc-block (Miltenyi) for 15 min at room temperature (RT), washed and stained with Live/Dead Fixable Near IR vital dye (ThermoFisher) for 15 min at RT in PBS prior to staining for 30 min at RT with a panel of monoclonal antibodies (mAbs) and tetramers including anti-human CD3 (BUV395, UCHT1, BD Pharmingen), CD4 (BUV496, SK3, BD Pharmingen), CD8α (BUV805, SK1, BD Pharmingen), CD26 (PE-Cy7, BA-5b, Biolegend), CD45 (AF700, 2D1, Biolegend), CD161 (BV650, HP-3G10, Biolegend), CD218α (APC, H44, Biolegend), TCRγδ (11F2, FITC, BD Pharmingen), TRAV1-2 (BV711; 3C10; Biolegend) and MR1-Ag tetramers (BV421, 1 μg/ml). Mouse cell suspensions were stained with viability dye 7-aminoactinomycin D (Sigma), the MR1 or CD1d tetramers and a panel of surface mAbs at RT for 30 min. For cocktails involving two tetramers, cells were subjected to avidin and biotin blocking (Dako Systems) between staining with each tetramer. Anti-mouse surface mAbs used include TCRβ (H57-597, BD Pharmingen), TCRγδ (FITC, GL3, eBioscience), B220 (BUV496, RA3-6B2, BD Pharmingen), CD3 (BV786, 145-2C11, BD Pharmingen), CD4 (BUV395, RM4-5, BD Pharmingen), CD8 (BUV805, 53-6.7, BD Pharmingen) and CD44 (AF700, IM7, eBioscience). Details of antibodies are summarised in Supplementary Table 6.

For single-cell sorting, cells were then washed twice and sorted on a BD FACS ARIAIII. For analysis, cells were washed twice, then fixed and permeabilised using a Foxp3 Fix/Perm Kit (eBiosciences) according to the manufacturer's instructions. Cells were then stained for PLZF (PE, Mags21F7, eBiosciences) for 30 min on ice. Finally, cells were washed twice prior to immediate acquisition on a BD LSR Fortessa equipped with a yellow-green laser.

**Tetramers**. Human MR1-5-OP-RU, human MR1-6-FP, mouse MR1-5-OP-RU and mouse MR1-Ac-6-FP were essentially produced as previously described[10,16]. In brief, truncated ectodomains of human MR1.C262S were expressed as inclusion bodies in *Escherichia coli* (strain BL21) along with human β2-microglobulin (β2m). MR1 and β2m inclusion bodies were then refolded in vitro in the presence or absence of 5-A-RU (produced in house, Fairlie Laboratory, University of Queensland) and methylglyoxal (Sigma), 6-FP or Ac-6-FP (Shirks laboratories) using oxidative refolding, prior to dialysis and subsequent purification using DEAE sepharose for human MR1.K43A (GE Healthcare) or Ni-NTA agarose for human MR1-5-OP-RU, human MR1-6-FP, mouse MR1-5-OP-RU and mouse MR1-Ac-6-FP (ThermoFisher). Human MR1.K43A was then further purified by size-exclusion chromatography and anion-exchange chromatography followed by chemical biotinylation prior to storage at −80 °C. Human MR1-5-OP-RU, human MR1-6-FP, mouse MR1-5-OP-RU and mouse MR1-6-FP were enzymatically biotinylated using BirA enzyme (produced in-house) followed by further purification by size-exclusion chromatography prior to storage at −80 °C. Human CD1d-α-GalCer tetramers were made in-house as previously described[3]. In brief, pHLSec vectors encoding the truncated ectodomain of human CD1d, and human β2m were expressed in HEK293S.N-acetylglucosaminyltransferase-I− (GnTI−) cells by transient transfection using PEI transfection reagent. Human CD1d monomers were then purified using Ni-NTA agarose, biotinylated with BirA enzyme and further purified by size-exclusion chromatography as above. Purified CD1d was loaded with α-GalCer (KRN7000, Avanti) at a 3:1 molar ratio overnight at RT. All monomers were tetramerised using streptavidin-BV421 (Biolegend).

**TCR transfection into HEK293T cells**. Transfections were performed as previously described[24]. In brief, HEK293T cells were co-transfected with pMIGII plasmids encoding full-length, p2a-linked TCR-α and TCR-β chains[42], along with a second pMIGII vector encoding p2a-linked human or mouse CD3εδγζ subunits, using Fugene6 transfection reagent (Promega) at 37 °C, 5% CO₂. After 3 days, cells were harvested and filtered through 100-μm filter mesh, washed with PBS and stained for 30 min at RT with a cocktail containing Live/Dead Fixable Near IR stain (ThermoFisher) plus MR1-Ag tetramers (PE), CD1d-α-GalCer tetramers (PE) or anti-human CD3 (PE, UCHT1, BD). Cells were then washed twice and immediately acquired on a BD LSR Fortessa equipped with a yellow-green laser.

**Flow cytometry**. All flow cytometric data were analysed using the Flowjo software (Treestar). For PBMC analysis, lymphocytes were gated using FSC-A and SSC-A, doublets excluded using FSC-A and FSC-H, viable lymphocytes gated as CD45+ Live/Dead-NIR− and αβ T cells were defined as CD3+ TCRγδ−. All plots of primary mouse samples are gated B220− lymphocytes after dead cell and doublet exclusion unless stated otherwise. HEK293T cell lines are also subjected to dead cell and doublet exclusion before selected for high GFP+-expressing cells. Flow cytometric gating strategy for mouse and human lymphocytes is shown in Supplementary Fig. 7.

**In vitro Ag stimulation of MAIT cells**. MR1-Ag monomers were diluted in PBS to graded concentrations of 10 or 1 μg/ml and coated onto 24-well flat-bottom plates, for 2 h at 37 °C. Plates were subsequently washed with PBS twice to remove unbound proteins. Splenocytes were then cultured for 5–7 days at 37 °C, in an incubator containing 5% $CO_2$. Cells were harvested at the end of culture and analysed via surface staining for expansion of MAIT cells.

**Intranasal Legionella infection model**. Legionella longbeachae strain NSW150 inoculums were prepared as described previously[2]. Briefly, bacterial cultures were grown to log-phase ($OD_{600}$ 0.2–0.6) in streptomycin-supplemented buffered yeast extract broth, for 16 h at 37 °C. Sufficient bacteria were quantitated by optical density (OD) measurements of 1 $OD_{600} = 5$ Å $\sim 10^8$ ml$^{-1}$, washed and diluted in PBS before delivery to mice. Mice were anaesthetised with isoflurane before intranasally inoculated with 50 μl of NSW150. Mice were then sacrificed after 7 days for organ collection.

**Single-cell TCR sequencing**. For human and mouse scTCRseq, T cells were stained as above and sorted at 1 cell/well into 96-well PCR plates (Eppendorf). cDNA was produced using the SuperScript VILO cDNA Synthesis Kit (Thermo-Fisher) before being subjected to two rounds of semi-nested multiplex PCR[30] using PCR master mix (ThermoFisher) and multiplexed human or mouse TCR primer sets as previously described in refs. [43] and [30]. Successfully amplified TCR genes as determined by agarose gel electrophoresis were subjected to Sanger Sequencing using internal TRAC or TRBC primers at Australian Genome Research Facility (AGRF), Melbourne. Sequence data were analysed using the IMGT V-QUEST sequence alignment software[44].

**TCR sequence logos**. Sequence logos were created using Seq2Logo web server[45] using an unclustered Shannon format with no pseudocounts. The size of each amino acid is proportional to its frequency. Amino acid colouring is based on side chain chemical properties; (red, acidic [DE]; blue, basic [HKR]; black, hydrophobic [ACFILMPVW]; green, neutral [GNQSTY].

**Reporting summary**. Further information on research design is available in the Nature Research Reporting Summary linked to this article.

## Data availability

The authors declare that the data supporting the findings of this study are available within the article, supplementary information files and Source data or are available upon reasonable requests to the authors.

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

## Acknowledgements
This work was supported by the National Health and Medical Research Council of Australia (NHMRC; 1063587, 1083942, 1113293, 1140126) and the Australian Research Council (ARC; CE140100011). H.-F.K. is supported by an NHMRC ECF fellowship (1160333); D.G.P. is supported by an NHMRC CDF fellowship (1144308); A.P.U. is supported by an ARC Future Fellowship (FT140100278); D.F. and D.I.G. are supported by NHMRC Senior Principal Research Fellowships (1117017 and 1117766, respectively). The authors thank Dr Marco Herold and Dr Andrew Kueh from Walter and Eliza Hall Institute and in association with the Australian Phenomics Network for generating and providing *Traj33* gene-deleted heterozygous mice; Zheng Ruan, Scott Reddiex, Marcin Ciula and Catarina Almeida for technical assistance and David Taylor, Kelsey Endler and the staff in the Doherty Institute Animal House for animal husbandry assistance. We also thank Tina Luke and staff at the Doherty Institute Flow Cytometry Facility and Vanta Jameson and staff at the Melbourne Brain Centre Flow Cytometry Facility for flow cytometry support.

## Author contributions
H.-F.K., N.A.G. and D.I.G. conceptualised and formulated the study and wrote the manuscript; H.-F.K., N.A.G., C.X., R.S., Z.Z., Z.C. and A.P.U. designed and performed experiments and interpreted results; D.F. provided key reagents; Z.C., D.F., J.M., D.G.P. and A.P.U. provided intellectual input and edited the manuscript.

## Additional information

**Competing interests:** Z.C., D.P.F. and J.M. declare that they are inventors on patents describing MR1 tetramers and MR1 ligands. The other authors declare no competing interests.

