## [Peer Review File · Nature Communications]

Reviewers' comments:

Reviewer #1 (NKT, MAIT)(Remarks to the Author):

MAIT cells recognize riboflavin biosynthetic intermediates presented by MR1, and they are generally described as having an invariant T cell receptor (TCR) alpha chain paired with somewhat more variable beta chains. The situation is more complex, however, as previous work from this group and several others showed that some human T cells with a MAIT antigen specificity do not express the invariant alpha chain. Furthermore, there are populations of MR1-reactive cells that respond to microbes lacking the riboflavin pathway or that are MR1 self-reactive. Here the authors extend this previous work and provide a very significant amount of additional details and data on MAIT cell heterogeneity.

In brief, some of the highlights are: 1. They have created TRAJ33 ko mice, and used this to characterize minor populations of mouse MAIT cells that use the canonical V region rearranged to J regions such as TRAJ9 or TRAJ12. These J segments preserve features of the conserved TRAJ33 rearrangement, such as a critical tyrosine at position 95. Additional tetramer binding cells in these TRAJ33 KO mice expressed different V alpha chains. TRAJ33 KO mice doubtless will prove useful as MAIT cell recipients in studies of gene function in transferred MAIT cells. 2. They find that the MAIT cells in the TRAJ33 ko mice that have different V alpha chains impart an altered specificity with reactivity to 5-OP-RU but also another antigen, Ac-6-FP presented by MR1. 3. A minority of human cells that bind 5-OP-RU MR1 tetramers lack expression of the canonical TRAV1-2 and this category includes cells from multiple donors cells that express TRAV36. Previously, an expanded population of these cells had been in one donor. 4. Some tetramer binding cells with non-canonical rearrangements have a MAIT-like phenotype defined by expression of several proteins including CD161 and CD218a, while other tetramer-binding cells lack high expression of these proteins. Interestingly, cells that lack CD161, etc., also do not express high amounts of PLZF, suggesting these lymphocytes would have had a different developmental pathway and function.

Because tetramer-binding mouse MAIT cells lacking the canonical TRAV1 V alpha region were not found in wild type mice, it is possible to question if these cells exist at all in wild type animals. Perhaps in wild type mice they are out competed by TRAV1 expressing cells. The authors should acknowledge this as a possibility. Regardless, it seems unlikely this tiny subset is biologically significant.

There is some confusion as to what constitutes a CD8 co-receptor. The homodimer of CD8 alpha chains does not provide co-stimulatory function, for example, it cannot positively select many class I restricted TCRs. Only the CD8 alpha-beta heterodimer is a co-stimulatory molecule, but unfortunately, the authors did not test for CD8 beta expression. CD8aa homodimers can be an activation/maturation marker, and the authors should be more careful in referring to "co-receptor" expression. It is unlikely, however, that the failure of the TRAV3-4/TRAJ40 cell line to bind to any MR1 tetramers is due to the absence of CD8—is there evidence that MR1 alpha 3 domain can interact with CD8?

Minor points:

Page 6. The authors state there is a trend toward lower tetramer+ cells in the TRAJ33 KO heterozygotes in the spleen and lung, but the figure indicates this is only significant in the spleen.

Page 7. The authors state they sequenced 45 paired TCR sequences, but in the same paragraph, they note 20 are TRAV1+ and 26 TRAV1- (=46)

Reviewer #2 (MAIT, microbial immunity)(Remarks to the Author):

The manuscript by Koay et al., is well written, and makes the point elegantly in the mouse model that the MAIT cells are largely dependent on TRAJ33. The data is well presented, with one concern, likely minor, that the authors should demonstrate in Figure 1 that the absence of TRAJ33 does not effect NKTs or other T cell subsets. The authors also demonstrate quite well that the "alternate" MAIT cells appear to be rare, and do not experience expansion. The mechanisms behind this are not fully addressed, but might simply be considered in the discussion i.e. is this dependent on antigen? The microbiome?

The human data is very intriguing, as the authors argue that MAIT are dominated by a "public" TCR that is 12aa in length, and contains the Tyr95 aa. Alternately, the authors argue that the TRAV1-2 negative cells contain less PLZF, and reflect as more diverse TCR. While the data as presented appear robust, there is a concern regarding the depth of the sequencing analysis, and the observation that there are a number of reports that do not fully conclude this definition of MAITs using a public TCR.

On the one hand, Greenaway et al have argued for the possibility of "public" TCRs for both MAITs and NKTs. This has also been supported by the work of Lepore. However, there have also been a been a diversity of TCRs published for MAIT cells that have a more diverse array of alpha chains, and the association of these alpha chains with functional recognition of both microbes and specific ligands, This would raise a concern regarding the depth of the sequencing efforts, or a functional distinction between cells that are simply tetramer positive vs those capable of recognizing ligand in the context of infection.

Conversely, there is published data that would suggest that tetramer positive MAIT cells that are not TRAV1-2+ would have a more classical phenotype with regard to PLZF, and other MAIT cell markers.

As a result, the discussion of the data could be far more nuanced.

Reviewer #3 (Ag/MHC/TCR interaction, repertoire)(Remarks to the Author):

Hui-Fern Koay et. al., Diverse MR1-restricted T cells in mice and humans.

In this manuscript, the authors generate TRAJ33 KO mice and analyze the development of the few remaining MAIT T cells in these mice. Further, novel populations of MAIT T cells are identified in humans that express TCRs with a broader TCR usage than the canonical, dominant clonotype.

Overall, this manuscript present some is a very interesting new data that is very well described in the context of the field. I really only have three questions (two technical) that should aid the overall manuscript.

In several points within the manuscript (including paragraph 1 of the discussion), the authors suggest that the TRAJ33 KO mice differ from MR-1 KO mice in the context of the development of the residue population of MR-1 reactive MAIT cells studied in this manuscript. Given the extremely infrequent development of these cells, it is formally possible that these cells arise from selection on an alternative MHC/MHC-like molecule. The authors should demonstrate (head to head analyses) that MR-1 KO mice do not carry these T cells.

In the original TRAJ-18 KO mice (not described here) the mice had defects in TCR rearrangement to TRAJs downstream of the genetic disruption. I may have missed it, but did the authors check whether overall TCRa rearrangement is fully intact in these mice? This point should be addressed, particularly

given the authors indication that these mice will be used in models of infection and autoimmunity to reveal the role MAIT cells.

In Fig 6C. Many of the TCR/pMHC pairs show a range of staining patterns, e.g., M33-64 TRAV1-2-TRAJ33 TRBV6-4 TCR binds many human and mouse MR1 ligands, and for the huMR1 6-FP, shows two populations (Tet Neg, Tet Pos). MAV36 TCRs also show mixed staining patterns. As presented, it is difficult to reconcile whether the mixed staining patterns is a product of non-specific "background" staining with some of the MR-1 tetramers, or is a product of weak affinity TCR/ligand binding combined with different expression levels of the TCR on the 293 cells. If staining is specific, one would expect to observed a direct correlation between TCR expression and tetramer binding.

Response to reviewers:

Reviewer #1 (NKT, MAIT)(Remarks to the Author):

MAIT cells recognize riboflavin biosynthetic intermediates presented by MR1, and they are generally described as having an invariant T cell receptor (TCR) alpha chain paired with somewhat more variable beta chains. The situation is more complex, however, as previous work from this group and several others showed that some human T cells with a MAIT antigen specificity do not express the invariant alpha chain. Furthermore, there are populations of MR1-reactive cells that respond to microbes lacking the riboflavin pathway or that are MR1 self-reactive. Here the authors extend this previous work and provide a very significant amount of additional details and data on MAIT cell heterogeneity.

In brief, some of the highlights are: 1. They have created TRAJ33 ko mice, and used this to characterize minor populations of mouse MAIT cells that use the canonical V region rearranged to J regions such as TRAJ9 or TRAJ12. These J segments preserve features of the conserved TRAJ33 rearrangement, such as a critical tyrosine at position 95. Additional tetramer binding cells in these TRAJ33 KO mice expressed different V alpha chains. TRAJ33 KO mice doubtless will prove useful as MAIT cell recipients in studies of gene function in transferred MAIT cells. 2. They find that the MAIT cells in the TRAJ33 ko mice that have different V alpha chains impart an altered specificity with reactivity to 5-OP-RU but also another antigen, Ac-6-FP presented by MR1. 3. A minority of human cells that bind 5-OP-RU MR1 tetramers lack expression of the canonical TRAV1-2 and this category includes cells from multiple donors cells that express TRAV36. Previously, an expanded population of these cells had been in one donor. 4. Some tetramer binding cells with non-canonical rearrangements have a MAIT-like phenotype defined by expression of several proteins including CD161 and CD218a, while other tetramer-binding cells lack high expression of these proteins. Interestingly, cells that lack CD161, etc., also do not express high amounts of PLZF, suggesting these lymphocytes would have had a different developmental pathway and function.

We thank the reviewer for their positive comments about our work.

Because tetramer-binding mouse MAIT cells lacking the canonical TRAV1 V alpha region were not found in wild type mice, it is possible to question if these cells exist at all in wild type animals. Perhaps in wild type mice they are out competed by TRAV1 expressing cells. The authors should acknowledge this as a possibility. Regardless, it seems unlikely this tiny subset is biologically significant.

While we did find MAIT cells lacking the canonical TRAV1 TRAJ33 TCR sequences in WT mice, albeit at low frequency (2/111), admittedly, we did not find any TRAV1 negative MAIT cells out of 111 WT MAIT cell clones tested. In order to directly test the biological relevance of TRAJ33 negative MAIT cells, we have now carried out *in vivo* microbial challenge experiments with a model of *Legionella longbeachae* infection, similar to our recent paper (Wang et al 2018 Nature Comms). In these experiments, we see that the TRAV1+ TRAJ33- MAIT cells expand in the lungs of these mice. Furthermore, a single, non-canonical, TRAV1 negative MR1-tetramer+ cell was detected in the WT mouse lung. We think the most likely explanation is that TRAJ33 negative MAIT cells do exist in WT mice, and they are functionally competent, but are relatively rare compared to the TRAJ33+ MAIT cells in most settings. These new experiments are now included as Figure 5 and in the results text (page 9-10) and additional discussion is provided (page 15).

There is some confusion as to what constitutes a CD8 co-receptor. The homodimer of CD8 alpha chains does not provide co-stimulatory function, for example, it cannot positively select many class I restricted TCRs. Only the CD8 alpha-beta heterodimer is a co-stimulatory molecule, but unfortunately, the authors did not test for CD8 beta expression. CD8aa homodimers can be an activation/maturation marker, and the authors should be more careful in referring to “co-receptor” expression. It is unlikely, however, that the failure of the TRAV3-4/TRAJ40 cell line to bind to any MR1 tetramers is due to the absence of CD8—is there evidence that MR1

alpha 3 domain can interact with CD8?

We thank the reviewer for raising this important point. We have carefully reworded reference to the nature of CD8aa versus CD8ab taking care not to refer to CD8 as a coreceptor in this case. With regard to the TRAV3-4 cell line, there is some precedent in the literature for CD8 playing a role in MAIT cell activation (Gold et al 2013, Mucosal Immunol; Kurioka et al. 2017. Frontiers Immunol), but admittedly there is no published evidence that CD8 can affect tetramer staining. We went back to check this cell line's original phenotype as these were index sorted and found that this clone was CD8 high and CD44low (in contrast to the other J α 33 KO MAIT cells, 96% of which were CD44+ (rebuttal figure 1)). It is also not uncommon when dealing with very rare cells to get an occasional contaminant within the sort gate, which may explain this clone but we included it to provide an honest picture of our results. In order to avoid over-speculation, we have removed the statement about the possible role for CD8 in the staining of these cells but explained that this was an immature CD44low phenotype. If the reviewers and/or editors prefer, we can remove this cell line from the figure and perhaps just mention that one of the lines failed to re-stain with MR1 tetramer, as data not shown.

Page 6. The authors state there is a trend toward lower tetramer+ cells in the TRAJ33 KO heterozygotes in the spleen and lung, but the figure indicates this is only significant in the spleen.

Apologies, this was an error during figure preparation as the significance indicator on the lung graph was left out. This data for % MAIT cells in lung was indeed significant and this has now been corrected in the figure.

Page 7. The authors state they sequenced 45 paired TCR sequences, but in the same paragraph, they note 20 are TRAV1+ and 26 TRAV1- (=46).

Many thanks for careful attention to these details and flagging this error which has now been corrected to a total of 46 sequences.

Reviewer #2 (MAIT, microbial immunity)(Remarks to the Author):

The manuscript by Koay et al., is well written, and makes the point elegantly in the mouse model that the MAIT cells are largely dependent on TRAJ33. The data is well presented, with one concern, likely minor, that the authors should demonstrate in Figure 1 that the absence of TRAJ33 does not effect NKTs or other T cell subsets.

We thank the reviewer for their positive comments and for raising this question. We have now included data demonstrating that TRAJ33 deficiency has no impact on NKT cells or gd T cell subsets, shown in supplementary figure 3 and mentioned in results text (page 6).

The authors also demonstrate quite well that the "alternate" MAIT cells appear to be rare, and do not experience expansion. The mechanisms behind this are not fully addressed, but might simply be considered in the discussion i.e. is this dependent on antigen? The microbiome?

We agree that the low precursor frequency in naïve wt mice may be due to a lack of expansion and possibly a product of a competitive niche. We have now carried out *in vivo* microbial challenge experiments with a model of *Legionella longbeachae* infection, similar to our recent paper (Wang et al 2018 Nature Communications). In these experiments, we see increased numbers of the non-classical TRAJ33⁻ MAIT cells in the lungs of TRAJ33 KO mice. We also show that the TRAJ33⁻ MR1-5-OP-RU tetramer⁺ cells from lungs of infected mice are PLZF⁺, similar to classical MAIT cells in WT mice and distinct from non-MAIT cells from WT and TRAJ33^{-/-} mice. These new experiments are now included as Figure 5 and in the results text (page 9-10) and additional discussion is provided (page 14-15).

Furthermore, we carried out additional TCR sequencing of WT and KO lung MR1 tetramer⁺ cells from these challenged mice. These new data show the expansion of canonical TRAV1 TRAJ33⁺ MAIT cells in WT lungs, and alternate TRAV1 TRAJ33⁻ cells in TRAJ33 KO lungs. Interestingly, isolated TRAV1 negative sequences were detected from both WT and TRAJ33 KO lungs of infected mice. This is now included in the results section (page 9-10), Figure 5D and additional discussion is provided (page 14-15).

The human data is very intriguing, as the authors argue that MAIT are dominated by a "public" TCR that is 12aa in length and contains the Tyr95 aa. Alternately, the authors argue that the TRAV1-2 negative cells contain less PLZF, and reflect as more diverse TCR. While the data as presented appear robust, there is a concern regarding the depth of the sequencing analysis, and the observation that there are a number of reports that do not fully conclude this definition of MAITs using a public TCR.

On the one hand, Greenaway et al have argued for the possibility of "public" TCRs for both MAITs and NKTs. This has also been supported by the work of Lepore. However, there have also been a diversity of TCRs published for MAIT cells that have a more diverse array of alpha chains, and the association of these alpha chains with functional recognition of both microbes and specific ligands, This would raise a concern regarding the depth of the sequencing efforts, or a functional distinction between cells that are simply tetramer positive vs those capable of recognizing ligand in the context of infection.

Conversely, there is published data that would suggest that tetramer positive MAIT cells that are not TRAV1-2+ would have a more classical phenotype with regard to PLZF, and other MAIT cell markers. As a result, the discussion of the data could be far more nuanced.

We acknowledge that TCR- α chain deep sequencing provides greater depth than single cell analysis. Indeed, we performed TCR deep sequencing of TRAV1-2⁺ MAIT cells in a recent paper (Gherardin et al 2018 Immunol. Cell Biol) and detected TRAJ12, TRAJ20, plus many other sequences at low frequency, but with this approach it is not possible to show they are truly MR1 reactive in the absence of TCR $\alpha\beta$ pairing.

In this paper, we chose to use single cell analysis for several reasons. Firstly, this provides important details about the matching of TCR- α and TCR- β chains. For example, using this approach we have shown that the TRAV36⁺ TCRs are highly enriched for pairing with TRBV28 – an observation we would not otherwise have made using a TCR- α chain deep sequencing approach. Secondly, knowledge of these matched TCR chains allows us to perform TCR transfer experiments to not only validate the MR1-reactivity of the isolated clones, but also to test the Ag-specificity of these TCRs. This, for example, has allowed us to show that the TRAV36⁺ population is specific to the microbial Ag, 5-OP-RU, which would not have been possible with deep sequencing. Finally, single cell analysis is ideal for rare populations such as the one studied here, whereas it would be difficult to isolate enough TRAV1-2-ve cells to really benefit from deep sequencing approaches. Accordingly, our approach of single cell analysis has allowed us to make important observations about this cell type that would otherwise not have been possible with TCR-deep sequencing. We nonetheless agree that the limited depth of our approach is an important point and we have included some discussion of this in our revised manuscript (page 16). We also acknowledge that there may be important functional differences between TRAV1-2⁺ and TRAV1-2⁻ cells. We had attempted to classify these into CD161⁺CD218⁺PLZF⁺ and CD161⁻CD218⁻PLZF⁻ subsets in our original submission, including a focus on the enrichment of novel invariant TRAV36⁺ TCRs, which are germline encoded, present in all donors tested and exclusively present in the 'MAIT-like' CD161^{HI}/CD218^{HI} population. We have also tried to ensure that the discussion is more nuanced with these various considerations in mind (page 16).

Reviewer #3 (Ag/MHC/TCR interaction, repertoire) (Remarks to the Author):

In this manuscript, the authors generate TRAJ33 KO mice and analyze the development of the few remaining

MAIT T cells in these mice. Further, novel populations of MAIT T cells are identified in humans that express TCRs with a broader TCR usage than the canonical, dominant clonotype.

Overall, this manuscript present some is a very interesting new data that is very well described in the context of the field. I really only have three questions (two technical) that should aid the overall manuscript.

We thank the reviewer for their kind and positive comments about our manuscript.

In several points within the manuscript (including paragraph 1 of the discussion), the authors suggest that the TRAJ33 KO mice differ from MR-1 KO mice in the context of the development of the residue population of MR-1 reactive MAIT cells studied in this manuscript. Given the extremely infrequent development of these cells, it is formally possible that these cells arise from selection on an alternative MHC/MHC-like molecule. The authors should demonstrate (head to head analyses) that MR-1 KO mice do not carry these T cells.

In contrast to TRAJ33 KO mice, we were unable to find a population of mature CD44⁺ MAIT-like cells in MR1 KO mice, even following enrichment for these cells (Supplementary figure 4). We have seen extremely rare MR1 tetramer⁺ events in the MR1 KO mice but these are so infrequent that it is difficult to know if these are really MR1-reactive, chance cross-reactivity after being selected by an alternative MHC/MHC-like molecule, or a result of rare non-specific staining. Furthermore, we have carried out in vivo microbial challenge experiments with a model of *Legionella longbeachae* infection, similar to our recent paper (Wang et al. 2018 Nature Comms). In these experiments, we included MR1 KO mice as controls and we did not detect any MR1 tetramer⁺ cells in the lungs of infected MR1 KO mice, in contrast to the TRAJ33KO mice. This supports the concept that, in contrast to TRAJ33KO mice, MR1 KO mice lack a functional population of MAIT or MAIT-like cells. These *Legionella* infection experiments and implications for this question raised by reviewer 3 are now included as Figure 5 and in the results text (page 9-10) and additional discussion is provided (page 14-15).

In the original TRAJ-18 KO mice (not described here) the mice had defects in TCR rearrangement to TRAJs downstream of the genetic disruption. I may have missed it, but did the authors check whether overall TCRa rearrangement is fully intact in these mice? This point should be addressed, particularly given the authors indication that these mice will be used in models of infection and autoimmunity to reveal the role MAIT cells.

This is a good point in light of the previous findings with TRAJ18 KO mice. In our original submission, we did report some TRAJ genes in the residual MAIT cells that were upstream of TRAJ33, but admittedly, this was a small sample. To directly address this, we have now carried out single cell TCR- α sequencing of 139 additional TRAJ33 KO CD4⁺CD8⁺ (DP) pre-selection thymocytes together with 92 WT DP thymocytes and have shown that in both cases, many of these clones use TRAJ genes upstream (TRAJ34-58) of TRAJ33. These data confirm that there is no significant difference in the frequency of TRAJ genes upstream of TRAJ33 in either immature TRAJ33KO DP thymocytes or residual TRAJ33 KO MAIT cells. These new data are included in Supplementary Figure 2 and in the results text (page 6).

In Fig 6C. Many of the TCR/pMHC pairs show a range of staining patterns, e.g., M33-64 TRAV1-2-TRAJ33 TRBV6-4 TCR binds many human and mouse MR1 ligands, and for the huMR1 6-FP, shows two populations (Tet Neg, Tet Pos). MAV36 TCRs also show mixed staining patterns. As presented, it is difficult to reconcile whether the mixed staining patterns is a product of non-specific "background" staining with some of the MR-1 tetramers, or is a product of weak affinity TCR/ligand binding combined with different expression levels of the TCR on the 293 cells. If staining is specific, one would expect to observed a direct correlation between TCR expression and tetramer binding.

Thank you for raising this point. The mixed staining is due to the range of GFP/CD3/TCR expression on the TCR transfected 293 cells. The reviewer is correct – there should be, and is, a direct correlation between TCR expression and tetramer binding. In our study, the amount of tetramer staining does indeed correlate with

the amount of GFP/TCR expression, but for some lower affinity interactions, the tetramer can only stain the cells with the highest amounts of TCR expression. Moreover, in order to avoid any potential steric hinderance between anti-CD3 antibodies and MR1 or CD1 tetramers, each transfected cell line was singly stained with antibodies or tetramers separately. For this reason, we applied a gate to each cell line to include only GFP/TCR bright cells, giving an MFI for CD3 of approximately 6000 for each cell line tested (as we have illustrated here in Rebuttal Figure 2, see right hand column showing CD3 staining). This allowed us to determine a GFP gate that provided this given CD3 level, and we could then apply the same GFP gate to tetramer-stained cells thereby standardising the TCR levels for each plot. This was the gating used to generate the histograms in Figure 6 (now Fig 7 in the revised version). Even with this GFP/TCR high gating, for some TCRs with weaker tetramer staining, reflecting lower affinity, only the highest GFP/TCR^{HI} cells within the GFP gate applied showed positive staining for tetramers (eg. as seen for the Hu MR1-6-FP tetramers in column 2). It is important to point out that we know this staining is specific because of our negative control tetramer (HuCD1d α -GalCer, fifth column and the 6-FP loaded MR1 tetramers in rows 1, 3 and 7). We had attempted to explain our approach to this gating in the figure legend of the original submission but have now more clearly explained this in the revised version. We prefer to show the histograms as per Figure 7 because they provide a simpler means to compare tetramer staining intensity but provide the data underlying these histograms for the reviewer to examine.

Rebuttal figure 1

Gating of index-sorted MR1-5-OP-RU tetramer+ cells from WT and *Tra33* KO α HSA-complement-depleted enriched thymus. Flow cytometry plots showing MR1-tetramer and TCR β ; CD44; and CD4 and CD8 expression profiles of sorted cells. Red boxes indicate corresponding expression of MR1-tetramer, TCR β , CD44, and CD8 for index sorted cell identified as TRAV3-4 TRAJ30.

**Rebuttal figure 2.**

Gating of 293t transient transfection. Flow cytometry plots showing MR1 (columns 1-5), CD1d (column 6) tetramer and CD3 (column 7) staining versus GFP on HEK293T cells transiently transfected to express human TCRs (left column). Cells are gated for viable 293T cells, MFI of gated CD3+ population is depicted on right of plots. Gates used to derive data in main manuscript figure 5 are shown on each graph.

References:

- Gold, M. C., T. Eid, S. Smyk-Pearson, Y. Eberling, G. M. Swarbrick, S. M. Langley, P. R. Streeter, D. A. Lewinsohn, and D. M. Lewinsohn. 2013. Human thymic MR1-restricted MAIT cells are innate pathogen-reactive effectors that adapt following thymic egress. *Mucosal immunology* 6: 35-44.
- Kurioka, A., A. S. Jahun, R. F. Hannaway, L. J. Walker, J. R. Fergusson, E. Sverremark-Ekstrom, A. J. Corbett, J. E. Ussher, C. B. Willberg, and P. Klenerman. 2017. Shared and Distinct Phenotypes and Functions of Human CD161⁺⁺ Valpha7.2⁺ T Cell Subsets. *Frontiers in immunology* 8: 1031.
- Wang, H., C. D'Souza, X. Y. Lim, L. Kostenko, T. J. Pediongco, S. B. G. Eckle, B. S. Meehan, M. Shi, N. Wang, S. Li, L. Liu, J. Y. W. Mak, D. P. Fairlie, Y. Iwakura, J. M. Gunnensen, A. W. Stent, D. I. Godfrey, J. Rossjohn, G. P. Westall, L. Kjer-Nielsen, R. A. Strugnell, J. McCluskey, A. J. Corbett, T. S. C. Hinks, and Z. Chen. 2018. MAIT cells protect against pulmonary *Legionella longbeachae* infection. *Nature communications* 9: 3350.
- Gherardin, N. A., M. N. T. Souter, H. F. Koay, K. M. Mangas, T. Seemann, T. P. Stinear, S. B. G. Eckle, S. P. Berzins, Y. d'Udekem, I. E. Konstantinov, D. P. Fairlie, D. S. Ritchie, P. J. Neeson, D. G. Pellicci, A. P. Uldrich, J. McCluskey, and D. I. Godfrey. 2018. Human blood MAIT cell subsets defined using MR1 tetramers. *Immunology and Cell Biology* 96: 507-525.

REVIEWERS' COMMENTS:

Reviewer #1 (Remarks to the Author):

I am in favor of removing the cell line from the figure and just stating that it did not re-stain with tetramer as data not shown.

Page 13—considering position 95 of the alpha chain, the authors imply that substitution of glycine for aspartic acid in TRAV37 TCRs is conservative, but this is not the case considering the loss of the negative charge.

Referee #2:

The authors have responded thoroughly to the critique.

Referee #3:

The authors have nicely addressed my concerns. Eric

Response to reviewers:

Referee #1:

I am in favor of removing the cell line from the figure and just stating that it did not re-stain with tetramer as data not shown.

We have moved the data from this cell line from the main figures to new supplementary figure 5 (in accordance with Nat Comms requirements to not list data as not shown).

Page 13—considering position 95 of the alpha chain, the authors imply that substitution of glycine for aspartic acid in TRAV37 TCRs is conservative, but this is not the case considering the loss of the negative charge.

We thank the reviewer for picking up this mistake. We have edited this section on page 14 accordingly.

Referee #2:

The authors have responded thoroughly to the critique.

Referee #3:

The authors have nicely addressed my concerns. Eric

We thank Reviewers 2 and 3 for their valuable input.